**Experimental observation of the impact of nanostructure on hygroscopicity and reactivity of fatty acid atmospheric aerosol proxies**

Adam Milsom,[1] Adam M. Squires,[2] Ben Laurence,[2] Ben Woden,[3] Andrew J. Smith,[4] Andrew D. Ward[5] and Christian Pfrang.[1,6,*]

[1]School of Geography, Earth and Environmental Sciences, University of Birmingham, Edgbaston, B15 2TT, Birmingham, UK.

[2]Department of Chemistry, University of Bath, South Building, Soldier Down Ln, Claverton Down, BA2 7AX, Bath, UK.

[3]Department of Chemistry, University of Reading, RG6 6AD, Reading, Berkshire, UK

[4]Diamond Light Source, Diamond House, Harwell Science and Innovation Campus, OX11 0DE, Didcot, UK.

[5]Central Laser Facility, STFC Rutherford Appleton Laboratory, Didcot OX11 0FA, UK

[6]Department of Meteorology, University of Reading, Whiteknights, Earley Gate, RG6 6BB, Reading, UK.

*Corresponding author: Professor Christian Pfrang (c.pfrang@bham.ac.uk)

**Abstract**

Atmospheric aerosol hygroscopicity and reactivity play key roles in determining an aerosol's fate and are strongly affected by its composition and physical properties. Fatty acids are surfactants commonly found in organic aerosol emissions. They form a wide range of different nanostructures dependent on water content and mixture composition. In this study we follow nano-structural changes in mixtures frequently found in urban organic aerosol emissions, i.e. oleic acid, sodium oleate and fructose, during humidity change and exposure to the atmospheric oxidant ozone. Addition of fructose altered the nanostructure by inducing molecular arrangements with increased surfactant-water interface curvature. Small-Angle X-ray Scattering (SAXS) was employed for the first time to derive the hygroscopicity of each nanostructure, thus addressing a current gap in knowledge by measuring time- and humidity-resolved changes in nano-structural parameters. We found that hygroscopicity is directly linked to the specific nanostructure and is dependent on the nanostructure geometry. Reaction with ozone revealed a clear nanostructure-reactivity trend, with notable differences between the individual nanostructures investigated. Simultaneous Raman microscopy complementing the SAXS studies revealed the persistence of oleic acid even after extensive oxidation. Our findings demonstrate that self-assembly of fatty acid nanostructures can significantly impact two key atmospheric aerosol processes: water uptake and chemical reactivity, thus directly affecting the atmospheric lifetime of these materials. This could have significant impacts on both urban air quality (e.g. protecting harmful urban emissions from atmospheric degradation and therefore enabling their long-range transport), and climate (e.g. affecting cloud formation), with implications for human health and wellbeing.

**Introduction**

Atmospheric aerosols represent a large uncertainty when considering their impact on the climate (Boucher et al., 2013; Shrivastava et al., 2017) and urban particulate matter makes a

significant contribution to air pollution, affecting air quality and health (Shrivastava et al., 2017;
Harrison, 2020; Chan and Yao, 2008; Pöschl, 2005). Organic matter can account for a large
portion of aerosol emissions depending on the emission source (Jimenez et al., 2009) and
environmental conditions have been shown to affect aerosol composition (Li et al., 2021).
There are both anthropogenic and biogenic sources of organic aerosols. Activities such as
cooking emit a range of organic compounds which can go on to form secondary organic
aerosol (SOA) (Zeng et al., 2020). Cooking emissions have been estimated to add *ca*. 10 %
to UK $PM_{2.5}$ emissions (Ots et al., 2016) and have been linked with poor air quality (Stavroulas
et al., 2024).

Oleic acid is a fatty acid and a common organic compound found in both cooking (Zeng et al.,
2020; Alves et al., 2020; Vincente et al., 2018) and marine emissions (Fu et al., 2013). It is
reactive towards common atmospheric oxidants such as ozone and $NO_3$, making it a model
compound for laboratory studies into aerosol properties (Zahardis and Petrucci, 2007;
Gallimore et al., 2017; Pfrang et al., 2017; Pfrang et al., 2011; Pfrang et al., 2010; King et al.,
2010; Sebastiani et al., 2022; Shiraiwa et al., 2012; Shiraiwa et al., 2010). Other common
organic emissions are saccharides (sugars), which are also found in urban (Wang et al., 2006)
and biogenic emissions (Fu et al., 2013; Fu et al., 2008; Kirpes et al., 2019). Sugar emissions
such as levoglucosan and glucose have been shown to react readily with Criegee
intermediates, which are formed during ozonolysis (Enami et al., 2017). The fact that these
two common classes of organic compounds (fatty acids and sugars) are found in the same
aerosol samples raises the possibility that they are able to interact; for example by a sugar
reacting with oleic acid ozonolysis Criegee intermediates, potentially altering the product
distribution and adding to the complexity of this reaction mechanism – a possibility explored
in this study.

Aerosol phase state has been predicted to vary significantly in the atmosphere and is linked
to factors such as composition, humidity and temperature (Shiraiwa et al., 2017; Schmedding
et al., 2020). One key influence on aerosol multiphase processes is particle viscosity (Reid et
al., 2018) and viscous phases have been identified by field measurements of SOA (Virtanen
et al., 2010). Particle viscosity can vary by orders of magnitude between phase states, which
means the diffusion coefficients of small molecules through the particle phase also vary and
heterogeneous processes (*i.e.* oxidation and water uptake) are affected (Shiraiwa et al., 2011;
Koop et al., 2011). Viscous phases can induce diffusion gradients during particle
humidification (Alpert et al., 2019; Hosny et al., 2016; Renbaum-Wolff et al., 2013; Zobrist et
al., 2011). Particles of oleic acid have also been observed to increase in viscosity as a result
of oxidation (Hosny et al., 2016). The fate of organic atmospheric aerosols is therefore strongly
influenced by their phase state.

Organic coatings are present on the surface of marine aerosols, where sugars and fatty acids
were found to be major constituents (Kirpes et al., 2019). Poor air quality has been linked to
high $PM_{2.5}$ surface organic content in Beijing, China (Zhao et al., 2020) and the long-range
transport of harmful substances emitted in the urban environment has been attributed to
viscous organic coatings and the phase state of the aerosol (Shrivastava et al., 2017; Mu et
al., 2018). Analysis of marine aerosols heavily influenced by anthropogenic activity found that
fatty acids were present along with Polycyclic Aromatic Hydrocarbons (PAHs) and phthalates,
which are known to cause poor health (Kang et al., 2017). There is a long-standing
discrepancy between the longer lifetime measured in the field compared to laboratory
measurements for oleic acid (Rudich et al., 2007; Wang and Yu, 2021). These observations
suggest that aerosols are able to travel far from their sources and that the formation of viscous
organic coatings could account for their long-range transport.

Pure oleic acid in the liquid phase exhibits some order by the formation of dimers (Iwahashi et al., 1991). As a surfactant, the addition of its ionic form (sodium oleate) and water can induce the formation of lyotropic liquid crystal (LLC) phases (Tiddy, 1980). These are three-dimensional nanostructures which can vary from spherical and cylindrical micelles to bicontinuous networks and bilayers. The spherical and cylindrical micelles can exist with "normal" (oil in water) or "inverse" (water in oil) curvature; the latter are the class formed by the systems in this paper (Pfrang et al., 2017). In our studies, the spherical inverse micelles can exist as (disordered) "inverse micellar" phases, or as ordered "close-packed inverse micellar" phases, which may have cubic (Fd3m) or hexagonal ($P6_3/mmc$) symmetry. The cylinders typically pack as hexagonal arrays ("inverse hexagonal phase") and the bilayers as "lamellar" stacks. These structures, shown in Figure 1, can be followed by Small-Angle X-ray Scattering (SAXS), which probes the nanometre scale. The close-packed inverse micellar, inverse hexagonal, and lamellar phases all show long-range periodicity, giving rise to Bragg peaks in SAXS patterns whose positions show symmetries and repeat spacings. The (disordered) inverse micellar phase gives a broad hump in SAXS, whose position shifts with micelle size.

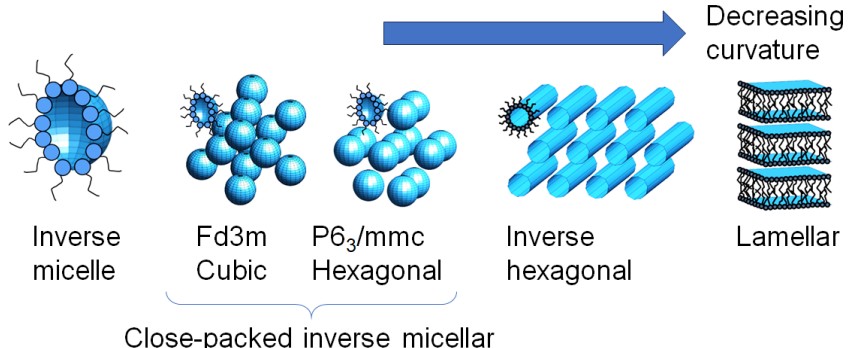

Figure 1: different phases formed by the surfactant systems in this study

Each of these structures exhibit varying physical properties, the key ones being diffusivity and viscosity. Diffusion coefficients can vary dramatically between micellar, close-packed micellar, inverse hexagonal and lamellar phases with diffusion in the latter two becoming directionally dependent (Lindblom and Orädd, 1994; Orädd et al., 1995). The diffusion of atmospherically relevant small molecules, such as ozone and water, would therefore also be affected by the nanostructure formed in the organic medium, affecting the key aerosol heterogeneous processes of water uptake and chemical reaction. While the present study is exploring the behaviour of organic aerosol components, we acknowledge the presence of other components in atmospheric aerosols, specifically inorganic species, which can undergo efflorescence and will add to the complexity of the behaviour of real atmospheric material compared to our organic-material focussed proxies.

We have previously demonstrated the feasibility of LLC formation in levitated particles of a fatty acid aerosol proxy (Pfrang et al., 2017; Milsom et al., 2023; Milsom et al., 2022a) and have exploited the SAXS experiment to quantify the effect of self-assembly on reaction kinetics (Milsom et al., 2021a), along with modelling of the potential impact on the atmospheric lifetime of LLC formation (Milsom et al., 2022b). In this study we coat capillaries with a self-assembled oleic acid/sodium oleate/fructose proxy use SAXS to follow changes in these nanostructures during humidity cycles and exposure to ozone. We investigate the sensitivity of the

nanostructure to proxy composition and humidity and demonstrate that reactivity is affected by nanostructure.

### Methods

**Preparation of self-assembled coatings inside quartz capillaries**

The method of film preparation is identical to that described in Milsom et al. (2021a). While
coatings inside quartz capillaries will only provide very limited insight on the behaviour of coatings on aqueous droplets (which are better approximated by floating self-assembled
monolayers at the air-water interface as in previous work, e.g. Pfrang et al., 2014, Woden et al., 2018 and Sebastiani et al., 2022), they are good proxies for coatings of solid particles in
the atmosphere such as mineral dust. Sample coating solutions were prepared as follows: oleic acid (Sigma-Aldrich, 90 %), sodium oleate (Sigma-Aldrich, 99 %) and fructose (Sigma-
Aldrich, 99 %) were dissolved as 10 wt % solutions in methanol and samples weighed to the desired ratio. All coating solutions are weighed as 1:1:x wt ratio mixtures (oleic acid:sodium
oleate:fructose), where x is 0.5, 1, 2 corresponding to 20, 33 and 50 wt % fructose compositions.
**SAXS experiment and simultaneous Raman microscopy on films coated inside quartz capillaries**
SAXS probes aggregates at the nanometre scale, measuring order at the molecular, rather than atomic (X-ray diffraction), scale (Li et al., 2016; Pauw, 2013). The scattered intensity is
measured against a scattering parameter ($q$) which is proportional to the scattering angle. $q$ is inversely proportional to the characteristic spacing between equivalent scattering planes ($d$)
via equation 1. This is also a measure of the spacings between inverse micelles.

$$d = \frac{2\pi}{q} \ (1)$$

This $d$-spacing can be used to determine a range of nano-structural parameters - for example, the water layer thickness between lamellar sheets (Kulkarni et al., 2011; Milsom et al., 2022c).
This experimental setup is the same as used in our previous capillary film study (Milsom et al., 2021a). Key experimental parameters are listed here: SAXS patterns were collected as 1s
exposures at different positions along the coated capillary with a delay of 75 s between each scan to avoid any X-ray beam damage; the beam size at the sample was approximately 320
x 400 µm (FWHM); SAXS patterns were acquired between $q$ = 0.008 – 0.6 Å$^{-1}$ by a *Pilatus P3-2M* detector.
The Raman microscopy setup is as described in Milsom et al. (2021a): A 532-nm Raman laser probe was focussed with a long working distance objective (numerical aperture: 0.42) and a
minimum spot diameter of ~ 1.5 µm. The emitted laser power was 20 – 50 mW. By following the oleic acid C=C bond peak at ~ 1650 cm$^{-1}$ and normalising to the –$CH_2$ peak at ~ 1442 cm$^{-1}$
, we were able to follow the progress of the ozonolysis reaction simultaneous to the SAXS measurements.
**Controlled humidification of coated films**

Humidity was monitored and controlled using a bespoke Raspberry Pi (RPi) system. Dry (room
air) and wet pumps were controlled by the RPi, in order to reach the target relative humidity (RH), which was measured by a sensor at the outlet of the coated capillary tube with a
precision of 2 %.

After samples were coated, they were left for ~ 15 min to equilibrate at room humidity (~ 50 –
60 % RH) before being attached to the humidity control system. The capillary was then
humidified to the desired settings using the RPi control programme adjusting humidity in the
range of ca. 40 to 90% RH.

**Ozonolysis of coated films**

The ozonolysis procedure follows what was set out previously (Milsom et al., 2021a) and is
summarised here: Oxygen (BOC, 99.5 %) was passed through a pen-ray ozoniser (Ultraviolet
Products Ltd., Cambridge, UK) which was calibrated offline by UV spectroscopy; the ozone
concentration for all ozonolysis experiments was 77 ± 5 ppm at a flow rate of 60 mL min$^{-1}$.
Note that such a high ozone concentration (atmospheric ozone levels rarely exceed 0.1 ppm)
was used as it is known that self-assembled semi-solid phases slow the rate of reaction
significantly (Pfrang et al., 2017; Milsom et al., 2021a). Therefore, comparatively high ozone
concentrations were chosen to be able to observe an oxidative decay during the limited
timescale of synchrotron experiments while they are substantially higher than those generally
encountered in the atmosphere. The ozone-oxygen mixture was measured to be at < 5 % RH.

Film thickness was determined by X-ray beam attenuation using diodes measuring the
incident and transmitted intensities. The maximum attenuation was determined by filling a
capillary with sample material. The thickness of each coated film was then calibrated by
comparison with the filled capillary's attenuation.


**Results and Discussion**

**Time- and humidity-resolved nanostructure changes**

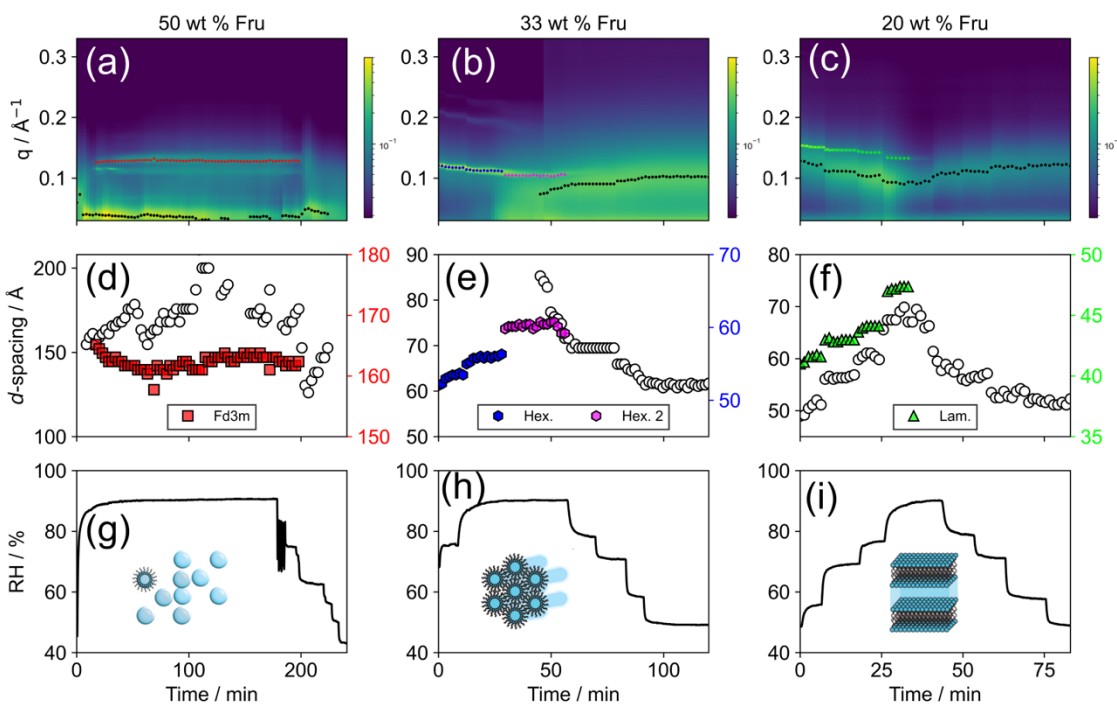

Figure 2. (a)-(c) SAXS patterns as a function of time during the humidity cycle. Peak positions
for inverse micellar (black markers) and specific nanostructures (coloured markers) are *q*
values corresponding to the time-resolved d-spacings represented in (d)-(f). (g)-(i)
Simultaneous RH vs time during the experiment. Each set of SAXS, *d*-spacing and RH data

is presented for each proxy fructose composition as wt % of organic mass with oleic acid and sodium oleate in a 1:1 wt ratio: (a), (d), (g) – 50 wt %; (b), (e), (h) – 33 wt %; (c), (f), (i) – 20 wt
% (*i.e.* 50 wt % fructose is a 1:1:2 oleic acid:sodium oleate:fructose mixture). The additional phases co-existing with the (disordered) inverse micellar phase are the cubic close-packed
inverse micellar (Fd3m) phase (a,d,g); two different inverse hexagonal phases (b,e,h); and the lamellar phase (c,f,i).
Different amounts of fructose in the organic mixture result in different self-assembled nanostructures (Fig. 2(a)-(c)). The inverse micellar phase is seen in all experiments, and this
co-exists with cubic close-packed inverse micellar, inverse hexagonal, and lamellar phases at 50 wt% fructose, 33 wt% fructose and 20 wt% fraction, respectively.  From first principles
fructose, as a hydrophilic water-soluble molecule, would be expected to facilitate water uptake into the organic phase and act as a humectant (moisture attracting agent), analogous to the
effect glycerol has on LLC phase boundaries (Richardson et al., 2015). By this logic, larger amounts of fructose should afford more hydrated phases at a given humidity. This can indeed
be seen from a comparison of the inverse micellar spacings at high relative humidity (Figures 2 & 3).  However, this does not explain the formation of a close-packed inverse micellar phase
at 50 wt% fructose vs. inverse hexagonal at 33 wt% fructose, and lamellar at 20 wt% fructose. We suggest that an additional effect is observed during our experiments: the water-surfactant
interfacial curvature increases with increasing fructose concentration (Figure 1). This is clear evidence for fructose acting as a *kosmotrope* – a water-structure-inducing molecule (Kulkarni
et al., 2011; Libster et al., 2008; Koynova et al., 1997). As a kosmotrope, fructose removes water from the water-surfactant interface. This reduces the effective surfactant headgroup
area, enabling the formation of structures with increased curvature at a given water content (in this case, experimental humidity – see corresponding cartoons of each phase in Fig. 1).
The phase boundary therefore shifts according to the amount of fructose in the mixture. A set of fructose content-dependent nanostructures are possible as a result. Each one of these
nanostructures possesses unique physical properties (as set out in the introduction). The sensitivity of the nanostructure to the amount of fructose in the system suggests that the
physical properties, which influence atmospheric trace gas uptake, could also change with similar sensitivity to aerosol composition.
The characteristic *d*-spacing for each of the observed nanostructures increases with increasing RH (Fig 2. (d)-(f)). This is the result of water filing the aqueous cavity in the inverse
LLC nanostructures observed here. The time and humidity-resolved SAXS patterns acquired in this study have allowed us to take advantage of this characteristic and observe subtle RH-
dependent changes in this parameter and directly measure the water uptake of a specific phase. This analysis can be applied to two coexistent phases, provided their SAXS peaks do
not overlap – as is the case in our study. The effect of these phases on water uptake is explored in the *Hygroscopicity of observed nanostructures* section.
The phase change observed when going from low to high RH is not reversible for the two organic compositions which initially formed inverse hexagonal and lamellar phases at < 90 %
RH (Fig. 2(b) and (c)). This suggests that the initial or final phases observed are meta-stable (e.g. the Fd3m and P6$_3$/mmc inverse micellar cubic phases can occur under the same
conditions for this system; Pfrang et al., 2017). Fig. 2(d–f) shows that for a given phase equilibrated with water vapour at a particular RH, the *d*-spacing is stable. This suggests that
these phases are in equilibrium, even if they are meta-stable.

For the 33 wt % fructose mixture, a second inverse hexagonal phase appears at high RH

before eventually transitioning to an inverse micellar phase (Fig. 2(b) and (e)). Indeed, between ~ 40 – 60 min the inverse micellar and hexagonal phase are observed simultaneously
in the mixture. There therefore is a heterogeneity in terms of molecular order and physical
properties associated with each of these nanostructures. This coexistent inverse micellar
phase is observed for all mixtures studied here.

The 50 wt % fructose mixture exhibits a reversible phase transition from inverse micellar to a
cubic close-packed inverse micellar (Fd3m) phase during a humidification-dehumidification
cycle (Fig. 2(a)). The Fd3m phase appeared only at the highest humidity setting (90 % RH).
The phase transition does not involve a significant change in phase topology, making the
transition more facile compared with the transition to an inverse hexagonal or lamellar phase
- although the Fd3m arrangement is thought to include inverse micelles of differing size
(Seddon et al., 1990; Shearman et al., 2010).


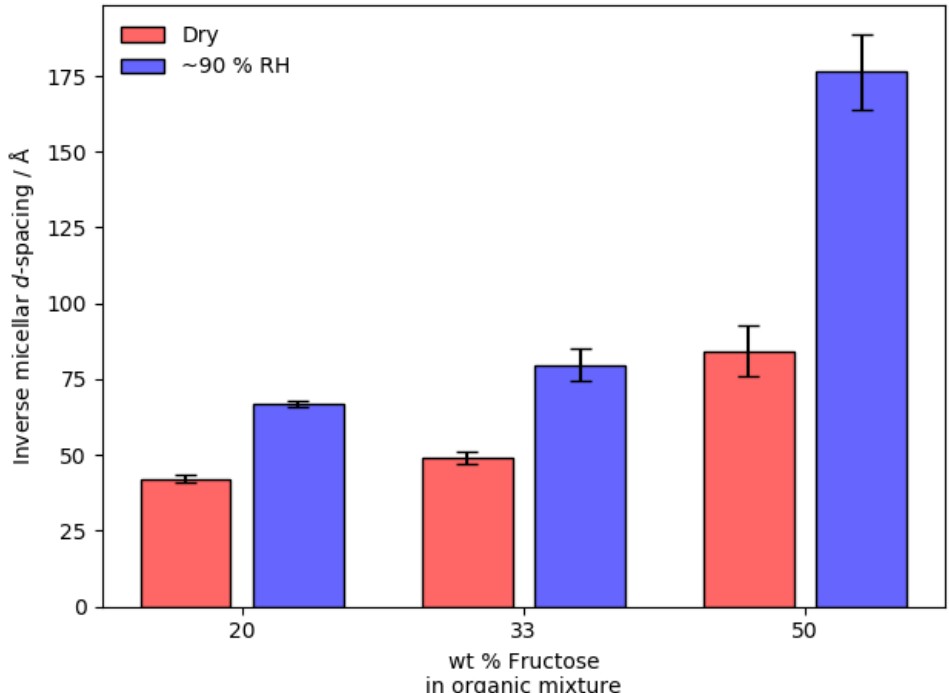

Figure 3. Inverse micellar *d*-spacing vs wt % fructose in the organic mixture under dry (~ 5 %
RH) and humid (~ 90 % RH) conditions. A clear increase in *d*-spacing is visible upon
humidification of each organic film.

A coexistent inverse micellar phase is observed for all organic compositions during these
humidity experiments (Fig. 2(a)-(c) – broad peak at lower *q* values). This coexistence
represents a heterogeneity within the organic film, implying a similar heterogeneity in physical
properties. We cannot say for certain whether this phase separation is uniform throughout the
film using this technique. However, the visible shift in the inverse micellar peak position during
humidity changes suggests that the change is happening in the majority of the film *i.e.* the
inverse micellar is distributed throughout the film.

The inverse micellar *d*-spacing increases with the amount of fructose in the mixture under dry
and humid conditions (Fig. 3). The inverse micellar phase observed for all fructose-containing
mixtures studied here has a much larger *d*-spacing than mixtures without fructose, where a *d*-
spacing of ~28-32 Å is expected (Fig. S1 – SAXS of a hydrated levitated particle of this
composition; Mele et al., 2018). Fructose therefore stabilises larger inverse micelles. Notably,
under dry (~5 % RH) conditions fructose seems to have a marked effect on the inverse micellar
*d*-spacing. This implies that fructose is collecting within the inverse micellar core and that

possibly some water has been accommodated within the structure, explaining the increase in the average repeat distance between inverse micelles.
Increasing the humidity substantially increases the inverse micellar *d*-spacing for all compositions. This effect is most potent for the 50 wt % fructose mixture (Fig. 3). The
observation highlights the ability of fructose to act as a humectant and stabilise large inverse micelles. It is worth restating that these inverse micellar phases at high RH are coexistent with
more ordered phases. A measure of the hygroscopicity for each coexistent phase can be extracted from the SAXS data and is presented in the *Hygroscopicity of observed*
*nanostructures* section.

**Hygroscopicity of observed nanostructures**


We have compared the hygroscopicity of the observed phases with what can be calculated

from Raoult's law for fructose over the RH range studied here.

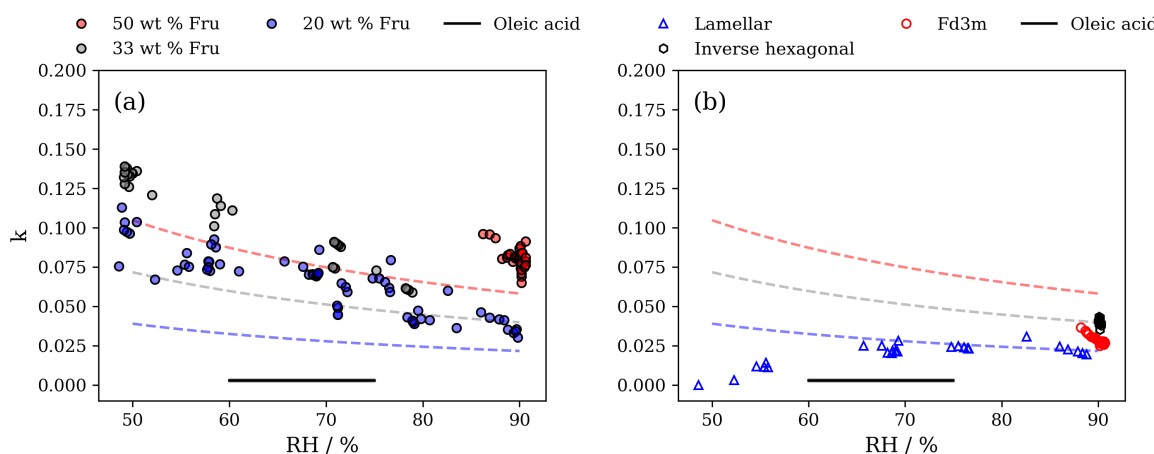

Figure 4. Plots of hygroscopicity parameter (*κ*) derived from *d*-spacings vs RH for the inverse

micellar phases at different weight percentage fructose (wt % Fru) (a) and for other nanostructures including the lamellar phase (at 20 wt % fructose), inverse hexagonal (at 33
wt % fructose) and close-packed inverse micellar (Fd3m, at 50 wt % fructose) (b). Dashed lines on both plots represent *κ* calculated for the same fructose-lipid ratio based on Raoult's
law at a particular RH. The colours of the dashed lines correspond to the wt % fructose in the mixture. The *κ* value for oleic acid measured by Rickards et al. (2013) is also plotted for
reference (*κ* = 0.003 ± 0.001).

κ-Köhler theory derives aerosol hygroscopicity from particle sizes at different water activities

($a_w$; Petters and Kreidenweis, 2007). The characteristic *d*-spacing calculated for each nanostructure observed here is related to its water content. We have applied κ-Köhler theory
by measuring the change in *d*-spacing with $a_w$, to describe the hygroscopicity of each phase. RH was converted to $a_w$ ($a_w$ = RH/100) and it is assumed that the proxy film had equilibrated
with the humidity inside the capillary (see the rapid change and equilibration of the *d*-spacing observed when changing RH in Fig. 2). Note that κ-Köhler theory is normally applied to aerosol
particles linking particle growth with humidity. Here we are not measuring individual particles, we are measuring nanoscale changes in the structural repeat distances, which are correlated
with water content. Equation 2 links the dry ($V_d$) and water ($V_w$) volumes with $a_w$ and a hygroscopicity parameter (*κ*; Petters and Kreidenweis, 2007).
$$\frac{V_w}{V_d} = \frac{a_w}{(1-a_w)}\kappa \quad (2)$$

The calculation of $\kappa$ is based on the geometry of each phase and the information regarding
the mass and volume fractions of the lipid and water regions derivable from the equilibrium $d$-
spacings obtained by SAXS (Asghar et al., 2015; Kulkarni et al., 2011). A detailed explanation
of the calculation of $\kappa$ is provided in the ESI.

This parameterisation of hygroscopicity is based on a simplified model which does not account
for non-ideal solution behaviour. Also, these experiments are not carried out on particle
ensembles or single particles, as has been the application previously (Liu et al., 2021;
Rickards, 2013). As theories of hygroscopicity are in general agreement at higher $a_w$ (RH)
(Rickards, 2013; Clegg et al., 1998; Wexler and Clegg, 2002; Fredenslund et al., 1975;
Topping et al., 2005; Zuend et al., 2008; Zuend et al., 2011), our measurements of $\kappa$ at high
RH (maximum 90 % RH) are the most informative. However, we caution the over-interpretation
of these $\kappa$ values in the context of other hygroscopicity studies due to the experimental
differences between this study and others. These $\kappa$ measurements do however provide a first
insight into the hygroscopic behaviour of these nanostructures and comparison between these
results is justified by the same method used to calculate $\kappa$.

The hygroscopicity of the disordered inverse micellar phase formed at each composition is
higher than what is predicted by Raoult's law for fructose (Fig. 4(a)). These predictions assume
that it is only the fructose that takes up water. Therefore, the formation of the inverse micellar
nanostructure, in addition to the hygroscopicity of the fructose, increases $\kappa$ beyond what would
be expected from the hygroscopicity of fructose alone.

The close-packed inverse micellar phase (Fd3m symmetry) appears to be less hygroscopic
than the Raoult prediction by a factor of ~ 2 at 90 % RH (Fig. 4(b)). This is in contrast to the
disordered inverse micelles coexistent with this nanostructure (Fig. 4(a)). The key difference
between the two nanostructures is that the close-packed inverse micelles are restricted in
space. The inverse hexagonal and lamellar phases are in better agreement with Raoult's law
predictions at > 85 % RH (Fig. 4(b)).

The lamellar phase appears to become much less hygroscopic at low RH. This may be
because of an increase in the inter-bilayer attractive forces at lower bilayer separations and/or
more restricted alkyl chains resulting from a more crystalline bilayer (Bahadur et al., 2019). A
crystalline form of this lamellar bilayer has been observed in similar systems (Tandon et al.,
2001; Milsom et al., 2021b).

As a thermodynamic parameter, $\kappa$ reflects the energy changes involved in changing the nano-
structural parameters associated with phase hydration and dehydration. For the lamellar
phase, work must be done in order to overcome inter- and intra-bilayer repulsion when
increasing and decreasing the volume of water between bilayers (Parsegian et al., 1979). To
clarify, if there is attraction between bilayers, then it is easier for the lamellar phase to lose
water (i.e. lower $\kappa$ at lower humidities where there is less distance and greater attraction
between bilayers). In the inverse hexagonal phase, the elastic free energy change associated
with a change in cylindrical radius is related to a bending modulus and the curvature of the
cylinder, both of which are associated with the bilayer-forming lipid and are affected by the
addition of other interacting molecules (Chen and Rand, 1997). The close-packed inverse
micellar phase is more sterically restricted than the disordered inverse micelles. The
disordered inverse micellar phase has the least frustrated hydrocarbon tails out of the
nanostructures presented here (*i.e.* they are not constrained close together, as is the case in
the inverse hexagonal and lamellar phases). Removing water from inverse micelles requires
more energy to do because of the increased curvature that results, explaining the increased

*κ* values for inverse micelles compared with the lamellar, inverse hexagonal and inverse cubic close-packed phases under similar conditions. These nanostructure-specific considerations help explain the difference between experiment and prediction.

All *κ* values derived from our SAXS data are greater than what has been measured for pure oleic acid (Fig. 4; Rickards et al., 2013). The addition of fructose alone does not account for all of the differences in *κ* observed between pure oleic acid, predictions based on Raoult's law and the nanostructured fatty acid proxy. There must be an effect of the nanostructure formed and this effect is most pronounced for the disordered inverse micellar phase. While the *κ* values reported here are substantially (up to nearly 50 times) above those previously measured for oleic acid (Rickards et al. (2013), it should be noted that *κ* values for highly-CCN-active salts such as sodium chloride are still higher (between 0.5 and 1.4; Petters and Kreidenweis, 2007), so that the inorganic fraction may be considerably more relevant than the nanostructure of the organic fraction for the potential of a particle to act as a CCN when considering internal mixtures of organic and inorganic materials in atmospheric particles.

**Reactivity-nanostructure relationship**

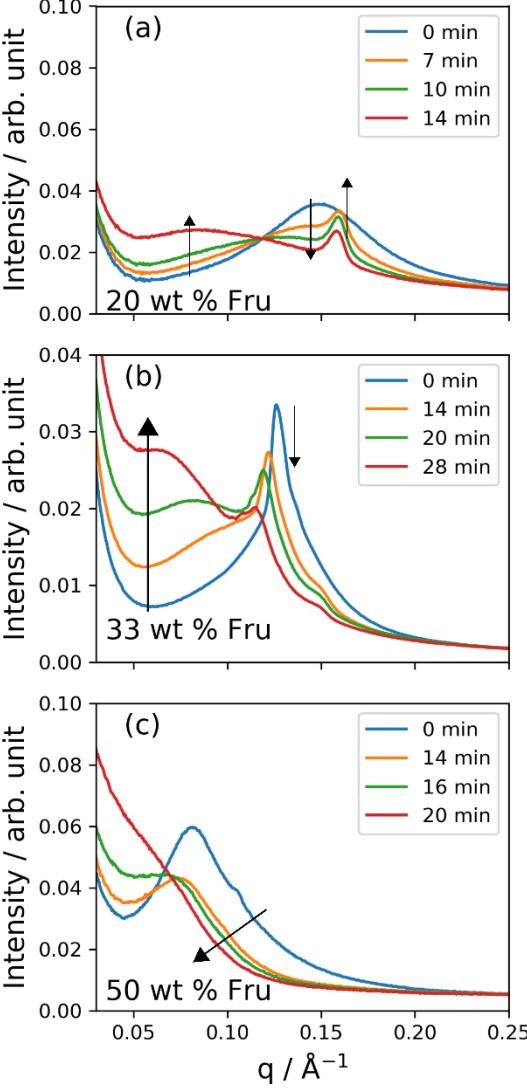

Figure 5. 1D SAXS patterns during ozonolysis of mixtures of: (a) 20 wt % fructose, (b) 33 wt % fructose and (c) 50 wt % fructose. Note the shift to low-$q$ of the broad inverse micellar peak for each composition. An additional phase appears in the first few minutes of reaction for the 20 wt % fructose mixture (a). The additional peaks associated with the ordered inverse micellar (P6$_3$/mmc) are revealed after ~ 20 min for the 33 wt % fructose mixture (b) – these are indexed in the ESI. [O$_3$] = 77 ± 5 ppm, RH < 5 %. The black arrows indicate the progression of different peaks from ordered phases with time as a visual guide.

We subjected proxy coatings of fatty acid-fructose mixtures to ozonolysis under dry conditions analogous to our previous film kinetic study (Milsom et al., 2021a). Figure 5 presents the SAXS patterns and phases observed during ozonolysis for the fructose-containing mixtures studied here. There are broad peaks characteristic of the inverse micellar phase in all mixtures, this was the most commonly observed phase under these conditions. An extra feature from an ordered phase appears during ozonolysis for the 20 wt % fructose mixture (Fig. 5(a)) – this is discussed in conjunction with simultaneous Raman spectrometry later (see Fig. 7). An ordered phase is observed for the 33 wt % fructose film (Fig. 5(b)). Initially, the less intense peaks associated with this phase are obscured by the broad overlapping inverse micellar peak. After ~ 20 min of ozonolysis the broad peak has shifted to lower $q$ and the other peaks are visible. These peaks index closest to a hexagonal close-packed inverse micellar phase with P6$_3$/mmc symmetry, which has been observed before in levitated droplets of a similar proxy (Pfrang et al., 2017) – see ESI for phase indexing. This allowed us to measure the kinetic difference between ordered and disordered inverse micellar phases.

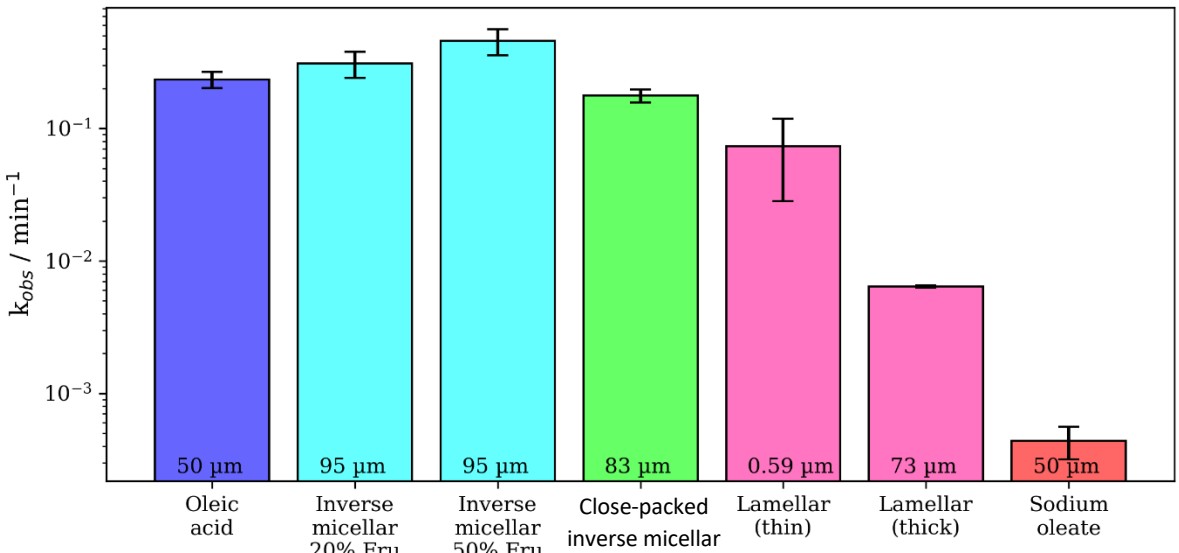

Figure 6. Pseudo-first order decay constants ($k_{obs}$) measured for the oleic acid-ozone reaction carried out on coated films of different composition and nanostructure. The thickness of each film is displayed at the bottom of each bar (see Table S1 in the ESI for all kinetic data and associated uncertainties). Oleic acid, sodium oleate and lamellar phase data are taken from earlier work (Milsom et al., 2021a). The lamellar phase was formed in a dry mixture of oleic acid: sodium oleate (1:1 wt). Oleic acid and sodium oleate decays were measured by following the C=C peak in the Raman spectrum as described in the methods. [O$_3$] = 77 ± 5 ppm and RH < 5 %.

Reaction kinetics can be followed by SAXS using an analysis technique that we have developed (Milsom et al., 2021a). We took advantage of the time resolution offered by a

synchrotron experiment to derive kinetic parameters for coated organic films of different composition and nanostructure (Fig. 6). All kinetic data are summarised in Table S1 and a more detailed derivation of these kinetic decay parameters is presented in Milsom et al. (2021a).

The disordered inverse micellar phase reacts faster than the ordered micellar phase coated at a similar thickness. This is to be expected as the close-packed inverse micelles are locked into their position, increasing the viscosity of the phase and therefore slowing the diffusion of small molecules such as ozone. The viscosity of close-packed inverse micelles can be in the order of $10^4$ times higher than for the disordered inverse micelles (Pouzot et al., 2007).

An order of reactivity exists between nanostructures. We are now able to compare the reactivity of different phases formed by this proxy system and the reactivity of its constituent parts (Fig. 6). Going from most to least reactive: inverse micellar > close-packed inverse micellar > (dry) lamellar. Note that the lamellar phase in this case is anhydrous. As suggested by Hearn *et al.*, diffusion of ozone past the closely packed lamellar chains is likely to be hindered and the rate of reaction reduced as a result, limiting the reaction to the surface of the film (Hearn et al., 2005).

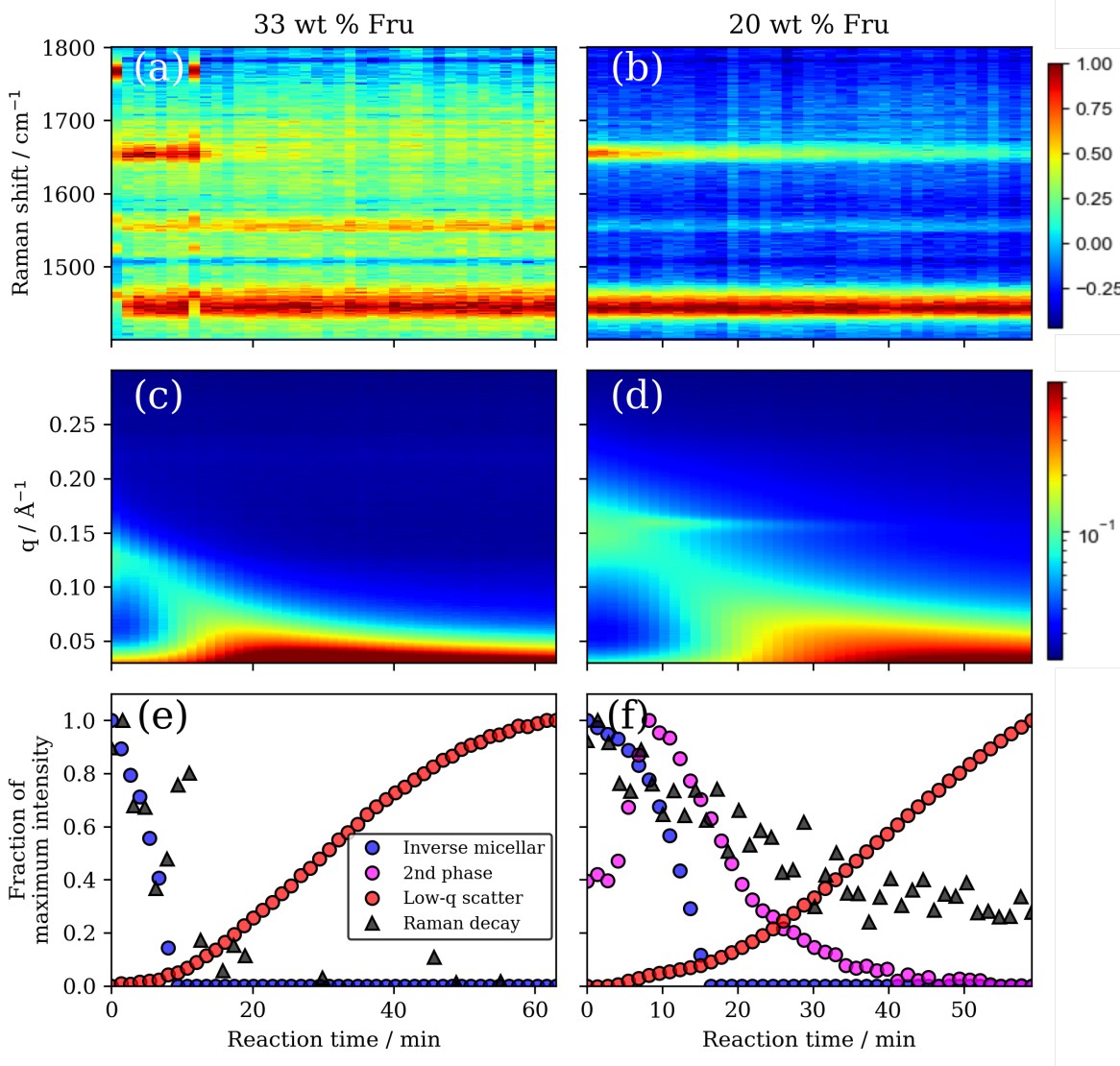

Figure 7. (a) and (b) Raman spectra vs reaction time highlighting the disappearance of the oleic acid C=C peak at ~ 1650 cm$^{-1}$ and the persistence of the –CH$_2$ deformation band at ~

$1442 \text{ cm}^{-1}$ for the 33 and 20 wt % fructose compositions, respectively. (c) and (d) simultaneous SAXS patterns vs reaction time showing the initial broad inverse micellar peak ($0.12$-$0.15 \text{ Å}^{-1}$)
which shifts to lower $q$ and disappears. The increase in low-$q$ scattering is also evident along with the appearance of a $2^{nd}$ phase peak for the 20 wt % fructose composition (d). (e) and (f)
Plots of maximum peak area intensity vs reaction time for key SAXS and Raman peaks. Raman decay is measured by following the $C=C$/-$CH_2$ peak area ratio.
The close-packed (ordered) inverse micellar phase film was ~ 12 µm thinner than the inverse micellar films. We have shown previously that film thickness can affect reactivity (Milsom et
al., 2021a), so we cannot rule out the effect of film thickness in these experiments. Though it was not possible to control film thickness, comparisons are still possible and actually reveal
some stark differences in reactivity. Most notably is the comparison of the sub-micron lamellar phase film with ~ 95 µm films of inverse micellar phase. The thin lamellar phase film reacts
slower than the inverse micellar films despite the ~ 160-fold difference in film thicknesses. There is also a difference of nearly two orders of magnitude in reactivity between the thickest
lamellar film (73 µm) and the inverse micellar films (95 µm).

     The inverse micellar $d$-spacing increases ($q$ decreases) as ozonolysis progresses (Fig. 7).

This experiment was carried out under dry conditions, so the increase in spacing must be a result of the reaction rather than any water uptake. We suggest that fructose itself reacts with
one of the intermediate products. Common saccharides found in the atmosphere, including glucose (closely related to fructose), have been shown to react readily with Criegee
intermediates that are formed as a result of ozonolysis (Enami et al., 2017). This forms ethers of greater mass and therefore products are likely to take up more space, accounting for the
increase in $d$-spacing observed during our ozonolysis experiment. Fructose can form an ether with oleic acid, however, to the author's knowledge, this has only been observed as an
enzymatic reaction (Ye and Hayes, 2011). Reaction with a Criegee intermediate is therefore the most probable explanation.
Products may themselves self-assemble. The increase in low-$q$ scattering observed here was not observed during reactions of similar samples without fructose (Milsom et al., 2021a). This
suggests that the species causing the increased low-$q$ scatter is associated with the fructose in the system. If high-molecular-weight fructose products are formed, the marked increase in
low-$q$ scatter suggests that these molecules aggregate into structures with large repeat distances.
A new phase was formed with a peak in $q$-range of ca. $0.14$–$0.16 \text{ Å}$ during the ozonolysis of the 20 wt % fructose mixture (Fig. 5(a) and Fig. 7(d)). This was unexpected as it was assumed
that self-assembly would be destroyed by chemical reaction of the constituent fatty acid, as observed previously (Pfrang et al., 2017; Milsom et al., 2021a). This phase took longer to
disappear compared with the initial inverse micellar phase. The reaction induced heterogeneity in the film both in the nanostructure and corresponding physical properties. This
observation suggests that there is a dynamic relationship between nanostructure and the chemical reaction of this fatty acid aerosol proxy. The identity of this phase is uncertain due to
the lack of a $2^{nd}$ order peak in the SAXS pattern, however this peak appears where the dry lamellar phase peak is expected to occur (Milsom et al., 2021a; Mele et al., 2018) – this is the
most likely arrangement. The atmospheric implications of the effect of nanostructure on reaction kinetics will be discussed in the following section.
There is evidence that the oleic acid double-bond persists at the end of the reaction (Fig. 7(b)). Simultaneous Raman spectroscopy on our deposited films shows clearly that the carbon-
carbon double-bond peak associated with oleic acid is still present at the end of the reaction even though the initial SAXS peaks are not visible. The increase in inverse micellar $d$-spacing

468 (SAXS peak shift to lower $q$), the notable increase in low-$q$ scattering and the persistence of the double-bond suggests that oleic acid may be protected by the increase in viscosity

470 expected by the formation of larger molecular mass molecules, which have been identified as products for the oleic acid-ozone system (Reynolds et al., 2006; Zahardis et al., 2005). This

472 persistence is consistent with most of the recent work on coated capillaries and residues observed after oxidising monolayers of atmospheric surfactants (including oleic acid) coated

474 on water (Milsom et al., 2021a; Woden et al., 2021; Woden et al., 2018; Sebastiani et al., 2022; Sebastiani et al., 2018; Pfrang et al., 2014; for completeness, it should be noted that

476 King et al., 2009, also reported a residue following oleic acid ozonolysis, although this finding was subsequently reported to be likely caused by an impurity in the deuterated sample used

478 in this early study and there was no evidence of such a residue in their most recent work, see King et al., 2020). This highlights the utility of a simultaneous technique to measure reaction

480 kinetics (Raman spectroscopy).

**Atmospheric implications**

482

 A wide distribution of aerosol phase states in the atmosphere has been observed and

484 predicted with global chemistry models (Shiraiwa et al., 2017; Schmedding et al., 2020; Virtanen et al., 2010). This phase state is dependent on the aerosol's environment, which

486 includes humidity and temperature. Aerosol multiphase processes are strongly affected by the formation of semi-solid and glassy phases due to reduced gas-particle interactions and the

488 effect on particle diffusivity (Berkemeier et al., 2016; Zhou et al. 2019; Zhou et al. 2013; Mikhailov et al., 2009: Koop et al., 2011; Zobrist et al., 2011). This in turn leads to phase-

490 dependent increases in aerosol atmospheric lifetimes and can facilitate the long-range transport of an aerosol substantially. Particle phase state and viscous aerosol organic coatings

492 have been linked to the long-range transport of polycyclic aromatic hydrocarbons (PAHs), which are particularly harmful to human health by acting as carcinogens (Shrivastava et al.,

494 2017; Mu et al., 2018).

 In the work presented here, we are adding a further organic aerosol component to our bottom-

496 up approach for this fatty acid aerosol proxy system with the addition of the sugar fructose, which is commonly found in urban and marine emissions. The addition of fructose induces

498 nano-structural changes by acting as a kosmotrope under humidified conditions. This shows that the nanostructure depends on the organic composition in addition to the relative humidity.

500 The presence of other aerosol components will likely impact the self-assembly reported here, but, we expect that fatty acid self-assembly still occurs in their presence as briefly outlined

502 below (compare discussion in Pfrang et al., 2017). Uncharged water-soluble components have been shown to dissolve in the aqueous region of the self-assembled structure, acting as a

504 humectant (in addition to the role as kosmotrope demonstrated for fructose in the present work) and allowing the self-assembly to occur at lower humidities. Charged water-soluble

506 inorganic components will have the same effect, but in addition, by changing the ionic strength and head group charge, will shift the phase boundaries between different self-assembled

508 structures. Hydrophobic aerosol components will partition into the non-aqueous regions of the self-assembled phases promoting the formation of inverse ('water-in-oil') phases.

510 In this study, we quantify two key properties affected by the nanostructure: hygroscopicity and reactivity. As illustrated in Fig. 4, the nanostructure increases the hygroscopicity parameter ($\kappa$)

512 by as much as a factor of ca. 10 to 50 compared to liquid oleic acid. Hygroscopicity determines the water uptake of aerosol at a specific RH; we have previously shown (Milsom et al., 2022a)

514 that aerosol water content strongly impacts on viscosity. Fig. 6 shows that the aerosol reactivity changes by nearly two orders of magnitude when altering the nano-structural

arrangement e.g. between a 73-µm thick lamellar film and a 95-µm thick inverse micellar film. This strong effect on aerosol reactivity associated with the nanostructure is likely due to
changes in viscosity and diffusivity. We acknowledge that the film thicknesses given in Fig. 6 are comparatively thick considering that most atmospheric aerosols accumulate in the 0.1–
2.5-µm range. However, as discussed in Pfrang et al. (2017), for thermodynamically equilibrated phases, no substantial size dependence is expected and we could confirm
consistent self-assembly from 500-nm films to 2-mm droplets, i.e. covering the key size range for atmospheric particles. If some of the phases identified in our atmospheric aerosol proxy
were not thermodynamically stable states, the exact phase observed at a given point in the experiment would depend on timescales and therefore droplet size/film thickness, but complex
self-assembly would still be expected to occur. In Milsom et al. (2021) we have reported the film thickness-dependent kinetic behaviour and measured the effect of the organic phase on
the kinetics.

  Previously, we have shown that ozonolysis destroys self-assembly in fatty acid aerosol proxies
(Pfrang et al., 2017; Milsom et al., 2021a). Here we additionally show that ozonolysis can induce the formation of a new intermediate molecular arrangement (see Fig. 7(d)),
demonstrating the possibility that self-assembly could be induced by the chemical reaction of these atmospheric molecules with ozone. This, in combination with humidity-induced phase
changes, suggests a dynamic aerosol phase state which is dependent on the molecular arrangement of the surfactant molecules.
Atmospheric aerosols exhibit heterogeneity both in terms of composition and physical properties (Kirpes et al., 2019; Schill et al., 2015). Particle viscosity can become
heterogeneous during chemical reaction and exposure to humidity (Hosny et al., 2016). We have now demonstrated that nano-structural heterogeneity exists during humidity change and
ozonolysis where different nanostructures coexist. There must therefore be a heterogeneity in hygroscopicity in our proxy films due to the link between nanostructure and κ (see Fig. 4). The
formation of an intermediate nanostructure during ozonolysis observed here suggests that viscosity may not be equal throughout the film and that the diffusivity of small molecules such
as ozone throughout the particle would also not be uniform, affecting the lifetime of the proxy (Shiraiwa et al, 2011b). The increase in *d*-spacing we observed between inverse micelles
during ozonolysis suggests that larger molecules are formed as a result of the reaction (see Fig. 5 and Fig. 7(c) & (d)). These larger molecules may also contribute to film heterogeneity
and alter the reactive lifetime of these molecules.

  Sugars and fatty acids, such as fructose and oleic acid, are commonly encountered
components of aerosols emitted in urban (Wang et al., 2006) and marine (Fu et al., 2013) environments. Specifically, saccharides (sugars) have been identified along with fatty acids
as major components of thick (µm-scale) organic coatings observed on sea spray aerosols (Kirpes et al., 2019) and also in a cafeteria environment (Alves et al., 2020), demonstrating
the wide range of environments our proxies represent. Their relative abundances can vary significantly depending on season, time of day and location. In this study we have shown that
the proxy sugar content has a substantial impact on aerosol physical properties via a change in nanostructure. We conclude that, as the relative amount of sugar and fatty acid changes
between environments, nanostructures could also vary depending on the location and emission type.
We have now demonstrated that the reactivity of surface-active oleic acid depends not only on whether it is self-assembled (Pfrang et al., 2017; Milsom et al., 2021a), but also on the
specific nanostructure it adopts (see Fig. 7). Our results suggest that the lifetime of surfactant material would depend on nanostructure, which in turn is linked to aerosol composition and
which is also affected by relative humidity. We would expect complex behaviour associated with humidity changes given that nanostructure both influences and is influenced by humidity.
The associated surfactant lifetime will also change. It should be noted that we have carried out the ozonolysis experiments presented here only at low humidity and at high ozone levels;
the possible implications of this deviation from atmospheric conditions would merit further investigation (noting experimental challenges associated with interfering reactions of highly
reactive OH radicals potentially produced in ozonolysis studies at high humidities). Our earlier modelling work (Milsom et al., 2022b) estimated significantly extended half-lives of
nanostructured (lamellar-phase) oleic acid for a range of atmospherically relevant film thicknesses and ozone levels (e.g. a half-life increase of ca. 10 days for a 0.75 μm film in ca.
25 ppb ozone; see Fig. 7 in Milsom et al., 2022b). The persistence of surface-active material has been demonstrated experimentally at the air-water interface (Woden et al., 2021; Woden
et al., 2018; Sebastiani et al., 2022; Sebastiani et al., 2018; Pfrang et al., 2014). Simultaneous Raman microscopy suggests that oleic acid can persist in the films studied here, a finding
consistent with non-fructose-containing films of this proxy (Milsom et al., 2021a). We have demonstrated that the reactive lifetime of oleic acid can vary by orders of magnitude as a result
of different molecular arrangements. There is a link between surfactant content and cloud droplet formation potential as a result of a reduction in surface tension (Bzdek et al., 2020;
Ovadnevaite et al., 2017; Facchini et al., 2000; Facchini et al., 1999). Therefore, any increase in surfactant lifetime would imply a similar increase of the cloud formation potential of a
surfactant-containing aerosol, such as aerosols emitted from cooking or sea spray containing oleic acid and/or related species.

## Conclusions

Our work has clearly shown that changes in the nanostructure, induced by humidity changes, can directly affect both water uptake and reactivity which are known to be two key aerosol
ageing processes.[e.g. Pöschl, 2005]

Crucially, we have demonstrated and quantified the direct link between the nanostructures
formed by fructose-containing fatty acid mixtures and the key aerosol properties of hygroscopicity and reactivity for the first time by utilising synchrotron SAXS and complimentary
Raman microscopy. This combination of SAXS and Raman data allowed us to infer key atmospheric aerosol properties and extract information from coexistent nanostructures to draw
comparisons between these. As a result, heterogeneity could be revealed during humidity exposure and ozonolysis. Our findings demonstrate that self-assembly of fatty acid
nanostructures can alter both water uptake and chemical reactivity. We have also shown that ozonolysis can induce the formation of a new intermediate molecular arrangement,
demonstrating the possibility that self-assembly could be induced by the chemical reaction of these atmospheric components with ozone. This, in combination with humidity-induced phase
changes, suggests a dynamic aerosol phase state which is dependent on the molecular arrangement of the surfactant molecules.
Our work demonstrated the fundamental effects of nanostructure on water uptake and reactivity. While these parameters in turn affect the particles´ impacts on air quality and
climate, a direct assessment of these effects is not within the scope of the work presented here.

## Data access statement

Data supporting with this study are available in the supporting information and from the corresponding author upon request.

**Conflicts of interest**

There are no conflicts to declare.

**Acknowledgements**

This work was carried out with the support of the Diamond Light Source (DLS), instrument I22 (proposal SM21663). AM wishes to acknowledge funding from NERC SCENARIO DTP award number NE/L002566/1 and CENTA DTP. The work was supported by NERC (research grant NE/T00732X/1). The authors would like to thank Nick Terrill (DLS), Tim Snow (DLS) and Lee Davidson (DLS) for technical support during beamtime experiments; Jacob Boswell is acknowledged for help at beamtimes. The authors are grateful to the Central Laser Facility for access to key equipment for the Raman work simultaneously to the DLS beamtime experiments.

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
