# Peer review of "Experimental observation of the impact of nanostructure on hygroscopicity and reactivity of fatty acid atmospheric aerosol proxies"

_EGUsphere, 2024_

## Referee Comment (RC1)

**General comments**

Milsom et al., present results from an X-ray scattering study of oleic acid/sodium oleate and fructose coated capillaries as a proxy for organic material present in atmospheric aerosol. The authors use SAXS to explore the nanostructure of different mixtures of oleic acid and fructose, of increasing fructose concentration. They investigate the effect of humidity on the specific nanostructures that are formed and calculate hygroscopicity parameters for the different structures comparing these values to those calculated using Raoult's law. They also investigate the effect of different nanostructures on reactivity during ozonolysis of oleic acid under dry conditions using SAXS and Raman microscopy. The authors conclude that humidity impacts the nanostructure in mixtures of oleic acid/sodium oleate and fructose and further show that specific nanostructures impact the rate of reaction for oleic acid ozonolysis. They link these findings back to urban air quality and climate stating that aerosol containing nanostructures can affect the degradation of harmful species present in aerosol, and the ability of aerosols to act as CCN.

The paper presents novel ideas and data interpretation regarding nanostructure formation in organic aerosols and is suitable for publication in ACP after addressing the comments below. The authors should further discuss the limitations of their study in relation to the complexity of atmospheric aerosol, the atmospheric relevance of the high ozone concentrations and low humidity conditions used during their ozonolysis experiments, and comment on how a coated quartz capillary tube relates to a coating on an aqueous droplet. Furthermore, to aid understanding, the authors should clearly state the different nanostructure types early in the paper and use the same terminology throughout.

**Specific comments**

- The authors should acknowledge the presence of other components in atmospheric aerosol, such as inorganics, which can undergo efflorescence, and discuss how this may impact their findings. This should be addressed in both the introduction and discussion sections to give a broader context to the relevance of their study.
- 2. Many of the film thicknesses given in Figure 5 are extremely thick considering the size range of atmospheric aerosols, and in particular CCN (~0.1 um diameter). The authors should therefore comment on the atmospheric relevance of such film thicknesses and associated decay constants.
- 3. The authors acknowledge that the ozone concentration they use in their ozonolysis experiments is high (ppm), compared to atmospheric concentrations (ppb). If atmospherically relevant ozone concentrations were used on films of these thicknesses, would a reaction occur on atmospherically relevant timescales? Please discuss.
- 4. a) Please explain why ozonolysis was performed under only dry conditions (<5 %), not representative of the majority of atmospheric conditions, especially in a marine environment (source of coated aerosols) and discuss how humidity could affect reaction rates. The authors should further discuss how their findings on a dry coating in a quartz tube can be related to a coating on an aqueous aerosol.
- b) In the discussion section on page 15, lines 506-508, the authors should directly mention that humidity influences specific nanostructures (as well as aerosol composition) and therefore also impacts the lifetime of a surfactant.

- c) Regarding the cloud formation potential of a surfactant containing aerosol discussed on page 15, lines 515 onwards, the authors should also acknowledge that the nanostructure will change at higher humidity (when cloud droplets form) and the associated surfactant lifetime will also change. Currently all reactivity information is based on ozonolysis under dry conditions (<5% RH).
- 5. a) Clearly label the nanostructures in Figure 1 and add an appropriate legend. It is very difficult to see the data points in Fig 1 a) to c) and their colours in relation to the data points in d) to f). Fd3m should be defined explicitly here, not later on Page 6, line 226. It would be very useful to the reader to refer to a larger schematic clearly showing each nanostructure in detail e.g. something like Fig 1 in Pfrang et al., 2017. This would also help with understanding when additional nanostructures are mentioned e.g. P63/mmc mentioned on page 6 and later regarding Figure 4.
- b) On page 5, line 185, the nanostructures should be clearly named at the start of the discussion.
- c) Related to the above, please be consistent with naming of all nanostructures throughout the paper. On page 5, line 199, it states that the nanostructures have different physical properties as outlined in the introduction, but the nanostructures discussed in Figure 1 and named in the introduction are somewhat different i.e. micellar compared to inverse micellar, hexagonal (cylindrical micellar) compared to inverse hexagonal etc. which can cause confusion for the reader. Sometimes ordered and disordered are mentioned and in other places these are omitted. The labels in the figures should be the same as the descriptions in the text e.g. page 11, Figure 5 and lines 381-382. Figure 5 shows inverse micellar and ordered micellar and the text discusses inverse micellar and close-packed inverse micellar which is the same as ordered micellar, as also stated in backets on page 12, line 386. It would be much clearer for the reader if the same terms were used consistently throughout.
- 6. On page 5, line 190, it is stated that an **additional** effect is observed from the presence of fructose. Please clarify whether it is indeed an additional effect, meaning it acts as both as humectant and kosmotrope or it is actually a different effect? Later in the paper on page 7, line 254 it states fructose acts as a humectant and then on page 14, line 462, it states that fructose acts as a kosmotrope.
- 7. The decay constants reported in Figure 5 show results from the current study (light blue and green bars) and previous studies (other coloured bars). The previous studies were conducted on films in the absence of fructose, which, when present, the authors postulate could react with Criegee intermediates and impact the kinetics. The authors also saw a lamellar phase in the presence of 20 wt % fructose so why is this not compared here instead of a lamellar phase in the absence of fructose? Please discuss.
- 8. What humidity range was studied in the humidification experiments? Although shown on x axes in Figure 1, this is not explicitly stated anywhere. The information should be added to the 'Controlled humidification of coated films' section.
- 9. Page 8, lines 284-285 state that a detailed description of the calculation of κ is given the in ESI but it is missing. There is only Table S2 stating the calculated values. Please add the description.

- 10. Please improve the presentation and description of the data in Figure 4 e.g. label specific peaks, e.g. ordered phase. There are several arrows present in the figure which are not mentioned in the figure caption, and it is difficult to follow the description of which peaks shift or appear in the text.
- 11. Please add a colour scale legend to Figure 6.
- 12. Why are no results shown in Figure 6 for the 50% wt Fructose concentration?
- 13. On page 14, lines 472-473 the sentence should make it clear that different nanostructures will cause different viscosities and therefore diffusivities. It initially reads as if there is something causing a change in viscosity (e.g. humidity) that effects the nanostructure but experiments were only conducted at < 5% RH so this is not the case.
- 14. The authors cite King et al., 2009 as a reference for material being left at the interface following oxidation of an oleic acid film at the air-water interface on page 13, line 443 and on page 15, lines 510-511. However, a more recent publication by King et al., PCCP, 2020 (The reaction of oleic acid monolayers with gas-phase ozone at the air water interface: the effect of sub-phase viscosity, and inert secondary components) showed this finding to be erroneous and due to impurities in the film material, concluding that when a pure oleic acid film is reacted, no material is left at the interface. Please correct or remove references.

**Minor comments**

Move the 'Controlled humidification of coated films' section on Page 4 before the 'Ozonolysis of coated films' section to mirror the order that the results are presented in.

Sentence on page 12, lines 393-394, specify that two orders of magnitude corresponds to the decay constant/reactivity as otherwise not clear.

The authors mention oleic acid and fructose can be found in the urban and marine environment in the introduction and go on to state that coatings are present on the surface of marine aerosols. Later in the discussion they refer to their proxy as relevant to the urban environment alone e.g. page 14, line 461. Please be consistent with the description.

Remove the word 'massively' from the conclusion on page 15, line 532. Experiments were conducted under a specific set of conditions and are not applicable to all scenarios, therefore the use of massively seems inappropriate.

**Technical corrections**

The first sentence in the abstract doesn't read well. Consider changing 'in determining **the** aerosol's fate' to 'in determining **an** aerosol's fate' or 'in determining **the fate of an** aerosol'

Page 2, line 82. Change ...in the urban environment **have** been... to ...in the urban environment **has** been...

Add the word 'section' to page 6, line 211 e.g. ...explored in the Hygroscopicity on observed nanostructures **section** and also to the end of line 257 on page 7, ...is presented in the Hygroscopicity on observed nanostructures **section**.

Page 13, line 420. Correct typo 'A new phase was formed in the with a peak...'

---

## Referee Comment (RC2)

In this study Milsom et al. investigate fatty acid/sugar mixtures using SAXS and Raman microscopy. In particular, films that are mixtures of oleic acid/sodium oleate and fructose are studied as proxies of atmospheric aerosols. The authors explore the different types of nanostructures formed within such internal mixtures as a function of fructose mass fraction and as a function of relative humidity. They also derive hygroscopicity values for the different nanostructures and investigate the reactivity of oleic acid with ozone within these. They find both hygroscopicity and reactivity to be impacted by the nanostructure of the mixtures.

Overall, I feel that the data presented here provides new insights into the physical chemical properties of atmospheric aerosols, a topic that is of interest for the readership of ACP. However, I would like the authors to address the points below, before publication in ACP.

**General comments:**

G1: Introduction, L91-112: Please elaborate on your introduction and discussion of the possible three-dimensional nanostructures. It would be good to introduce the nanostructures relevant to this study already here along with e.g. a schematic (as you have them already as very small insets in your Fig. 1). This would greatly help the reader to visualize and distinguish the various nanostructures discussed throughout the text. I also encourage the authors to add clear labels and terms to each of the nanostructures and use the same terminology throughout the text, to make it easier for a reader to follow.

G2: Ozonolysis of coated films, L148: The ozone concentrations used herein are not atmospherically relevant, as acknowledged by the authors. It would be good to reiterate this when discussing Fig. 4 and the timescales therein. Given typical tropospheric ozone concentrations are much lower, what would be the typical atmospheric timescales at which the changes in SAXS pattern take place? In this regard, it would be helpful to consider discussing timescales of ozone exposure, to take the ozone concentration into account.

G3: The authors studied three different mixtures of oleic acid: sodium oleate:fructose, different in their mixing ratios, or simpler fructose fractions. I am missing some information whether these ratios were chosen to cover typical mixing ratio ranges found in the atmosphere, or for any specific other reasons? Furthermore, the authors note on L198 that "a set of fructose content-dependent nanostructures are possible". Neglecting the RH-dependency of (irreversible; L212) phase changes, can the authors comment on the range of fructose fractions for which they expect each of the three nanostructure types depicted in Fig. 1 (Fd3m phases, hexagonal phases, lamellar phases) to be present? I.e. is the Fd3m structure always expected in such mixtures if the fructose mass fraction is larger than 50% wt?

G4: Deriving hygroscopocity values for the different nanostructures is an interesting approach. The $\kappa$-values derived are extremely low, indicating a low water uptake capacity of theses nanostructures. For inorganic materials $\kappa$-values are often between 0.5 to 1.4, i.e. significantly higher than those observed here. The authors should comment on whether subtle differences in hygroscopicity as observed here (Fig. 3) would still be resolvable in aerosols that are internal mixtures of organic and inorganic material, as is often the case for atmospheric particles? Related, for the potential of a particle to act as a CCN, the inorganic fraction is probably considerably more relevant than the nanostructure of the organic fraction. I encourage the authors to include a discussion on this in their Section "Hygroscopicity of observed nanostructures" and when discussing the impact of nanostructure on hygroscopicity (L466-467) and cloud formation (e.g. L534-535) and climate.

**Specific comments:**

L75: Please add: DOI: 10.1039/c9cp03731d

L116: Add punctuation: "in Milsom et al. (2021a)."

L117-119: Can you comment if and to what degree esterification of the oleic acid and the methanol can take place during sample preparation, and how this could impact your results?

L120: "oleic acid: sod**i**um oleate:fructose)"

Fig. 1a-c: The meaning of the background colormap is unclear; not specified. Also, consider using non-filled, open markers for "inverse micellar" structures in panel d-f, to allow for easier visual distinction.

L188: add "(water-absorbing substance)" or "(moisture attracting agent)" or similar to clarify the meaning of humectant.

L189: The relation between "more hydrated" and "lower water-surfactant interactant interfacial curvature" warrants some more detailed explanation considering the discussion following in this paragraph.

L212-214: Is the statement of the irreversibility of the phase change generally true, or could this be dependent on the drying rate, when going from high back to low RH?

L220-222: I might be missing something here, but I do not see the coexistence of pink and blue hexagons between 40-60 min in Fig. 1e that you seem to refer to here in the text.

L227: Your Fig. 1d shows red squares (i.e. Fd3m structures) for ~175-200 min, i.e. at RH < 90%, please clarify.

L236: This inverse micellar phase does only seem to appear after around 50 min in Fig. 1e, despite the RH being constant for the previous ~20 min. Why is that?

L297: Add: "The *hygroscopicity of the* disordered…"

Fig. 4: Figure 4 and the discussion of it (L344 onwards) warrant some improvements. I interpret the solid black arrows in Fig. 4 as indicators of general trends, which peaks disappear and appear, but a clear explanation is missing. Consider adding some specific labels to the individual peaks. Also, the same-colored line corresponds to different times during ozonolysis across the different panels, which is a bit misleading. Having a consistent color code (map) throughout the panels could help with that.

L382: I am unclear why you specify "(dry) lamellar" here, since your ozonolysis was only performed under dry conditions (L344). Related, would you expect the stated trend to be different at different RH conditions?

Fig. 6: a-d: An explanation of the color map(s) is missing and should be added. Also, adding horizontal lines to highlight the peaks at 1650 $cm^{-1}$ and 1442 $cm^{-1}$ in panels a and b could help to guide the eye.

L475-480: This is a very interesting finding in my eyes. Is there a way to highlight this aspect already in the abstract of the manuscript? Right now, it is only mentioned here and on L426.

L532: "… with significant impacts on air quality and climate": I suggest toning this down a little bit here, as your work shows the fundamental effects of nanostructure on water uptake and reactivity. These parameters in turn affect the particles´ impacts on air quality and climate, but an assessment of these effects is not done here.

---

## Author Comment (AC1)

**Reviewer 1**

We are very grateful for the reviewer's positive assessment of our submission. We have addressed the constructive and helpful comments in detail below that have improved our manuscript significantly in our view. Our comments are added in **bold** font and new text is indicated in *italics*.

**"General comments**

Milsom et al., present results from an X-ray scattering study of oleic acid/sodium oleate and fructose coated capillaries as a proxy for organic material present in atmospheric aerosol. The authors use SAXS to explore the nanostructure of different mixtures of oleic acid and fructose, of increasing fructose concentration. They investigate the effect of humidity on the specific nanostructures that are formed and calculate hygroscopicity parameters for the different structures comparing these values to those calculated using Raoult's law. They also investigate the effect of different nanostructures on reactivity during ozonolysis of oleic acid under dry conditions using SAXS and Raman microscopy. The authors conclude that humidity impacts the nanostructure in mixtures of oleic acid/sodium oleate and fructose and further show that specific nanostructures impact the rate of reaction for oleic acid ozonolysis. They link these findings back to urban air quality and climate stating that aerosol containing nanostructures can affect the degradation of harmful species present in aerosol, and the ability of aerosols to act as CCN.

The paper presents novel ideas and data interpretation regarding nanostructure formation in organic aerosols and is suitable for publication in ACP after addressing the comments below."

**We are grateful for the positive assessment of our work in terms of suitability for publication in ACP and have addressed the reviewer's comments line-by-line below.**

"The authors should further discuss the limitations of their study in relation to the complexity of atmospheric aerosol, the atmospheric relevance of the high ozone concentrations and low humidity conditions used during their ozonolysis experiments, and comment on how a coated quartz capillary tube relates to a coating on an aqueous droplet."

We have added further details on the limitations of our approach considering the complexity of real atmospheric aerosols compared to our fairly simple proxies, emphasised the fact that our ozone concentrations are above those found in the atmosphere, consider more carefully humidity impacts and commented on the limitations of the quartz capillary approach to represent coatings on droplets. The coating is not a proxy for coating on aqueous aerosol, but for coating of solid particles e.g. mineral dust; in addition the experiments can give fundamental insights into the behaviour of the materials we have studied.

We have added new text in the methods section: "Therefore, comparatively high ozone concentrations were chosen to be able to observe an oxidative decay during the limited timescale of synchrotron experiments while they are substantially higher than those generally encountered in the atmosphere." and "(atmospheric ozone levels rarely exceed 0.1 ppm)" and ". While coatings inside quartz capillaries will only provide very limited insight on the behaviour of coatings on aqueous droplets (which are better approximated by floating self-assembled monolayers at the air-water interface as in previous work, e.g. Pfrang et al., 2014, Woden et al., 2018 and Sebastiani et al., 2022), they are good proxies for coatings of solid particles in the atmosphere such as mineral dust."

We have also added new text in the atmospheric implications section: "and which is also affected by relative humidity. It should be noted that we have carried out the ozonolysis experiments presented here only at low humidity and at high ozone levels; the possible implications of this deviation from atmospheric conditions would merit further investigation."

"Furthermore, to aid understanding, the authors should clearly state the different nanostructure types early in the paper and use the same terminology throughout."

In response to the reviewer's request, we have added a schematic diagram (new Figure 1) and more introductory detail to clarify which are the different nanostructures of interest and carefully checked for clear terminology throughout. We have added the following text (p3, from line 95) and we have ensured that we keep consistent nomenclature throughout:

"The spherical and cylindrical micelles can exist with "normal" (oil in water) or "inverse" (water in oil) curvature; the latter are the class formed by the systems in this paper (Pfrang et al., 2017). In our studies, the spherical inverse micelles can exist as (disordered) "inverse micellar" phases, or as ordered "close-packed inverse micellar" phases, which may have cubic (Fd3m) or hexagonal (P63/mmc) symmetry. The cylinders typically pack as hexagonal arrays ("inverse hexagonal phase") and the bilayers as "lamellar" stacks. These structures, shown in Figure 1, can be followed by Small-Angle X-ray Scattering (SAXS), which probes the nanometre scale. The close-packed inverse micellar, inverse hexagonal, and lamellar phases all show longrange periodicity, giving rise to Bragg peaks in SAXS patterns whose positions show symmetries and repeat spacings. The (disordered) inverse micellar phase gives a broad hump in SAXS, whose position shifts with micelle size."

**New Figure:**

Close-packed inverse micellar

*"Figure 1: different phases formed by the surfactant systems in this study"*

**"Specific comments**

1. The authors should acknowledge the presence of other components in atmospheric aerosol, such as inorganics, which can undergo efflorescence, and discuss how this may impact their findings. This should be addressed in both the introduction and discussion sections to give a broader context to the relevance of their study."

We have taken the reviewer's valid point on board and amended the text of the revised manuscript accordingly. We have added the following text in the introduction: "While the present study is exploring the behaviour of organic aerosol components, we acknowledge the presence of other components in atmospheric aerosols, specifically inorganic species, which can undergo efflorescence and will add to the complexity of the behaviour of real atmospheric material compared to our organic-material focussed proxies."

We also added a brief discussion how these inorganic compounds and other aerosol components may impact our findings by adding "The presence of other aerosol components will likely impact the self-assembly reported here, but, we expect that fatty acid self-assembly still occurs in their presence as briefly outlined below (compare discussion in Pfrang et al., 2017). Uncharged water-soluble components have been shown to dissolve in the aqueous region of the self-assembled structure, acting as a humectant (in addition to the role as kosmotrope demonstrated for fructose in the present work) and allowing the self-assembly to occur at lower humidities. Charged water-soluble inorganic components will have the same effect, but in addition, by changing the ionic strength and head group charge, will shift the phase boundaries between different self-assembled structures. Hydrophobic aerosol components will partition into the non-aqueous regions of the self-assembled phases promoting the formation of inverse ('water-in-oil') phases."

"2. Many of the film thicknesses given in Figure 5 are extremely thick considering the size range of atmospheric aerosols, and in particular CCN (~0.1 um diameter). The authors should therefore comment on the atmospheric relevance of such film thicknesses and associated decay constants."

We have taken the reviewer's valid point on board and amended the text of the revised manuscript accordingly. We have added the following text in the atmospheric implications section: "We acknowledge that the film thicknesses given in Fig. 6 are comparatively thick considering that most atmospheric aerosols accumulate in the 0.1-2.5-µm range. However, as discussed in Pfrang et al. (2017), for thermodynamically equilibrated phases, no substantial size dependence is expected and we could confirm consistent self-assembly from 500-nm films to 2-mm droplets, i.e. covering the key size range for atmospheric particles. If some of the phases identified in our atmospheric aerosol proxy were not thermodynamically stable states, the exact phase observed at a given point in the experiment would depend on timescales and therefore droplet size/film thickness, but complex self-assembly would still be expected to occur. In Milsom et al. (2021) we have reported the film thickness-dependent kinetic behaviour and measured the effect of the organic phase on the kinetics."

3. The authors acknowledge that the ozone concentration they use in their ozonolysis experiments is high (ppm), compared to atmospheric concentrations (ppb). If atmospherically relevant ozone concentrations were used on films of these thicknesses, would a reaction occur on atmospherically relevant timescales? Please discuss."

We have added a comment in the atmospheric implications section discussing this aspect with reference to our model study (Milsom et al., ACP, 2022) where we specifically modelled the impact of nanostructured oleic acid ozonolysis in the atmospherically relevant  $[O_3]$  range of 0-150 ppb: "Our earlier modelling work (Milsom et al., 2022b) estimated significantly extended half-lives of nanostructured (lamellar-phase) oleic acid for a range of atmospherically relevant film thicknesses and ozone levels (e.g. a half-life increase of ca. 10 days for a 0.75  $\mu$ m film in ca. 25 ppb ozone; see Fig. 7 in Milsom et al., 2022b)."

"4. a) Please explain why ozonolysis was performed under **only** dry conditions (<5 %), not representative of the majority of atmospheric conditions, especially in a marine environment (source of coated aerosols) and discuss how humidity could affect reaction rates. The authors should further discuss how their findings on a dry coating in a quartz tube can be related to a coating on an aqueous aerosol."

This is a valid point. As commented earlier, the coated capillaries are not considered a good proxy of aqueous aerosols, but for coated solid particles; we have carried out a wide range of studies of fatty acid monolayers at the air-water interface previously, addressing aqueous aerosols and have cited this work throughout the present study using coated capillaries; we have added the following **text outlined earlier in the method section to clarify this** *"While coatings inside quartz capillaries will only provide very limited insight on the behaviour of coatings on aqueous droplets (which are better approximated by floating self-assembled monolayers at the air-water interface as in previous work, e.g. Pfrang et al., 2014, Woden et al., 2018 and Sebastiani et al., 2022), they are good proxies for coatings of solid particles in the atmosphere such as mineral dust."*

Coated aerosols are indeed formed from marine sea spay, but another important source especially in densely populated urban areas are cooking emissions. This was already stated in the introduction, so we did not feel the need to add new text: "Oleic acid is a fatty acid and a common organic compound found in both cooking (Zeng et al., 2020; Alves et al., 2020; Vincente et al., 2018) and marine emissions (Fu et al., 2013)."

While we fully agree that ozonolysis studies at high RH would be very interesting, there are significant experimental challenges associated with high RH ozonolysis studies, specifically the fact that highly reactive OH radicals would be formed in our experimental system and the interpretation of our findings would be much less clear as multiple highly reactive species would be present and the short time scale of beamtime experiments would not allow adequate deconvolution of these complex processes. We added a comment spelling this out specifically: "It should be noted that we have carried out the ozonolysis experiments presented here only at low humidity and at high ozone levels; the possible implications of this deviation from atmospheric conditions would merit further investigation (noting experimental challenges associated with interfering reactions of highly reactive OH radicals potentially produced in ozonolysis studies at high humidities)."

"b) In the discussion section on page 15, lines 506-508, the authors should directly mention that humidity influences specific nanostructures (as well as aerosol composition) and therefore also impacts the lifetime of a surfactant."

We have added the underlined text to the end of the sentence (p15 line 522) so it now reads "Our results suggest that the lifetime of surfactant material would depend on nanostructure, which in turn is linked to aerosol composition and which is also affected by relative humidity."

"c) Regarding the cloud formation potential of a surfactant containing aerosol discussed on page 15, lines 515 onwards, the authors should also acknowledge that the nanostructure will change at higher humidity (when cloud droplets form) and the associated surfactant lifetime will also change. Currently all reactivity information is based on ozonolysis under dry conditions (<5% RH)."

We appreciate that high RH associated with cloud formation would change the nanostructure; indeed, we would expect complex behaviour given that nanostructure both influences and is influenced by humidity; in this paper we focussed on dry ozonolysis for experimental reasons as outlined above. To address the reviewer's comment, we have added the following text to the atmospheric implications section in the revised manuscript: "We would expect complex behaviour associated with humidity changes given that nanostructure both influences and is influenced by humidity. The associated surfactant lifetime will also change."

"5. a) Clearly label the nanostructures in Figure 1 and add an appropriate legend. It is very difficult to see the data points in Fig 1 a) to c) and their colours in relation to the data points in d) to f). Fd3m should be defined explicitly here, not later on Page 6, line 226. **It would be very useful to the reader to refer to a larger schematic clearly showing each nanostructure in detail** e.g. something like Fig 1 in Pfrang et al., 2017. This would also help with understanding when additional nanostructures are mentioned e.g. P63/mmc mentioned on page 6 and later regarding Figure 4."

We have added Fd3m to the text added to the introduction where the phases are introduced together with a new Figure 1 showing each nanostructure in detail as proposed by the reviewer (see page 3, line 95). We have also added the following text into the figure caption (see page 6, line 215): "The additional phases co-existing with the (disordered) inverse micellar phase are the cubic close-packed inverse micellar (Fd3m) phase (a,d,g); two different inverse hexagonal phases (b,e,h); and the lamellar phase (c,f,i)."

"b) On page 5, line 185, the nanostructures should be clearly named at the start of the discussion."

**The following new text was added at the start of the discussion (page 6, line 219):** "Different amounts of fructose in the organic mixture result in different self-assembled nanostructures (Fig. 2(a)-(c)). The inverse micellar phase is seen in all experiments, and this co-exists with cubic close-packed inverse micellar, inverse hexagonal, and lamellar phases at 50 wt% fructose, 33 wt% fructose and 20 wt% fraction, respectively."

"c) Related to the above, **please be consistent with naming of all nanostructures throughout the paper**. On page 5, line 199, it states that the nanostructures have different physical properties as outlined in the introduction, but the nanostructures discussed in Figure 1 and named in the introduction are somewhat different i.e. micellar compared to inverse micellar, hexagonal (cylindrical micellar) compared to inverse hexagonal etc. which can cause confusion for the reader. Sometimes ordered and disordered are mentioned and in other places these are omitted. The labels in the figures should be the same as the descriptions in the text e.g. page 11, Figure 5 and lines 381-382. Figure 5 shows inverse micellar and **ordered micellar** and the text discusses inverse micellar and **close-packed inverse micellar** which is the same as ordered micellar, as also stated in backets on page 12, line 386. It would be much clearer for the reader if the same terms were used consistently throughout."

**We are very grateful for the reviewer's careful reading of the terminologies; we have now ensured we use "inverse hexagonal" and "inverse close-packed micellar" throughout.**

"6. On page 5, line 190, it is stated that an **additional** effect is observed from the presence of fructose. Please clarify whether it is indeed an additional effect, meaning it acts as both as humectant and kosmotrope or it is actually a different effect? Later in the paper on page 7, line 254 it states fructose acts as a humectant and then on page 14, line 462, it states that fructose acts as a kosmotrope."

We thank the reviewer for bringing to our attention the lack of clarity in this part of our argument. We did indeed mean that fructose acts both as a humectant and a kosmotrope, with the two aspects accounting for two different observed trends within the data. We have re-written the relevant paragraph on page 6 to say *"From first principles fructose, as a hydrophilic water-soluble molecule, would be expected to facilitate water uptake into the organic phase and act as a humectant (moisture attracting agent), analogous to the effect glycerol has on LLC phase boundaries (Richardson et al., 2015). By this logic, larger amounts of fructose should afford more hydrated phases at a given humidity. This can indeed be seen from a comparison of the inverse micellar spacings at high relative humidity (Figures 2 & 3). However, this does not explain the formation of a close-packed inverse micellar phase at 50 wt% fructose vs. inverse hexagonal at 33 wt% fructose, and lamellar at 20 wt% fructose. We suggest that an additional effect is observed during our experiments: the water-surfactant interfacial curvature increases with increasing fructose concentration (Figure 1)."*

"7. The decay constants reported in Figure 5 show results from the current study (light blue and green bars) and previous studies (other coloured bars). The previous studies were conducted on

films in the absence of fructose, which, when present, the authors postulate could react with Criegee intermediates and impact the kinetics. The authors also saw a lamellar phase in the presence of 20 wt % fructose so why is this not compared here instead of a lamellar phase in the absence of fructose? Please discuss."

We thank the reviewer for the comment; the lamellar phase disappeared during drying and we only conducted ozonolysis in dry conditions (as justified above), so we could not collect equivalent data for oxidation of a lamellar phase; we added the following wording in the atmospheric implications section in explanation: "We would expect complex behaviour associated with humidity changes given that nanostructure both influences and is influenced by humidity. The associated surfactant lifetime will also change. It should be noted that we have carried out the ozonolysis experiments presented here only at low humidity and at high ozone levels; the possible implications of this deviation from atmospheric conditions would merit further investigation (noting experimental challenges associated with interfering reactions of highly reactive OH radicals potentially produced in ozonolysis studies at high humidities)."

"8. What humidity range was studied in the humidification experiments? Although shown on x axes in Figure 1, this is not explicitly stated anywhere. The information should be added to the 'Controlled humidification of coated films' section."

The RH is displayed in Fig. 2, but we have now also added the range in the Methods section adding: *"adjusting humidity in the range of ca. 40 to 90% RH."*

"9. Page 8, lines 284-285 state that a detailed description of the calculation of  $\kappa$  is given the in ESI but it is missing. There is only Table S2 stating the calculated values. Please add the description."

We are grateful for the comment from the reviewer; the submitted version of the supporting information was indeed missing that information and we have now added a new section S4 in the new supporting information document providing a detailed description of the calculation of  $\kappa$  and have moved Table S2 to the end of this section.

"10. Please improve the presentation and description of the data in Figure 4 e.g. label specific peaks, e.g. ordered phase. There are several arrows present in the figure which are not mentioned in the figure caption, and it is difficult to follow the description of which peaks shift or appear in the text."

Thanks for the comments; we have added the following clarifying text in the figure caption: "The black arrows indicate the progression of different peaks from ordered phases with time as a visual guide."; all peaks are from ordered phases.

"11. Please add a colour scale legend to Figure 6."

**We have added a colour scale legend as requested by the reviewer in the revised Fig. 7 (these are arbitrary units).**

"12. Why are no results shown in Figure 6 for the 50% wt Fructose concentration?"

As these challenging experiments require highly competitive beamtime access at a large-scale facility, we had to be very selective when choosing the range of conditions to study; the kinetics of the composition of 50% wt fructose could not be run within these constraints.

"13. On page 14, lines 472-473 the sentence should make it clear that different nanostructures will cause different viscosities and therefore diffusivities. It initially reads as if there is something causing a change in viscosity (e.g. humidity) that effects the nanostructure but experiments were only conducted at < 5% RH so this is not the case."

**We have amended the wording to now read (line 579 in the revised manuscript) "This strong effect on aerosol reactivity associated with the nanostructure" to clarify this point.**

"14. The authors cite King et al., 2009 as a reference for material being left at the interface following oxidation of an oleic acid film at the air-water interface on page 13, line 443 and on page 15, lines 510-511. However, a more recent publication by King et al., PCCP, 2020 (The reaction of oleic acid monolayers with gas-phase ozone at the air water interface: the effect of sub-phase viscosity, and inert secondary components) showed this finding to be erroneous and due to impurities in the film material, concluding that when a pure oleic acid film is reacted, no material is left at the interface. Please correct or remove references."

We apologise for the inclusion of King et al. (2009) among the list of papers providing evidence for material left at the air-water interface – the reviewer is entirely correct with the comment that the conclusions in this particular study were affected by an impurity in the reactant used as discussed in King et al. (2020) – we have removed King et al. (2009) from the list of six papers providing evidence and amended the text (see underlined text below) highlighting this point made by the reviewer: "This persistence is consistent with most of the recent work on coated capillaries and residues observed after oxidising monolayers of atmospheric surfactants (including oleic acid) coated on water (Milsom et al., 2021a; Woden et al., 2021; Woden et al., 2018; Sebastiani et al., 2022; Sebastiani et al., 2018; Pfrang et al., 2014; for completeness, it should be noted that King et al., 2009, also reported a residue following oleic acid ozonolysis, although this finding was subsequently reported to be likely caused by an impurity in the deuterated sample used in this early study and there was no evidence of such a residue in their most recent work, see King et al., 2020)." We have also removed the reference to King et al. (2009) in the atmospheric implications section and added the most recent paper (King et al., 2020) to the list of references.

**"Minor comments**

Move the 'Controlled humidification of coated films' section on Page 4 before the 'Ozonolysis of coated films' section to mirror the order that the results are presented in."

**We have implemented this change in the revised manuscript.**

"Sentence on page 12, lines 393-394, specify that two orders of magnitude corresponds to the decay constant/reactivity as otherwise not clear."

**We inserted "in reactivity" to clarify this point.**

"The authors mention oleic acid and fructose can be found in the urban and marine environment in the introduction and go on to state that coatings are present on the surface of marine aerosols. Later in the discussion they refer to their proxy as relevant to the urban environment alone e.g. page 14, line 461. Please be consistent with the description."

**We added "and marine" to address this point.**

"Remove the word 'massively' from the conclusion on page 15, line 532. Experiments were conducted under a specific set of conditions and are not applicable to all scenarios, therefore the use of massively seems inappropriate."

**We have removed the term "massively" as requested.**

**"Technical corrections**

The first sentence in the abstract doesn't read well. Consider changing 'in determining the aerosol's

fate' to 'in determining **an** aerosol's fate' or 'in determining **the fate of an** aerosol' Page 2, line 82. Change ...in the urban environment **have** been... to ...in the urban environment **has**

been..."

**We have amended the text following these helpful corrections.**

"Add the word 'section' to page 6, line 211 e.g. ...explored in the Hygroscopicity on observed nanostructures **section** and also to the end of line 257 on page 7, ...is presented in the Hygroscopicity on observed nanostructures **section**."

**We have amended the text following these helpful corrections.**

"Page 13, line 420. Correct typo 'A new phase was formed in the with a peak..."

**Thanks for identifying this typo; we have corrected the text to read** *"A new phase was formed with a peak"*.

**Reviewer 2:**

We are very grateful for the reviewer's positive assessment of our submission. We have addressed the constructive and helpful comments in detail below that have improved our manuscript significantly in our view. Our comments are added in **bold** font and new text is indicated in *italics*.

"In this study Milsom et al. investigate fatty acid/sugar mixtures using SAXS and Raman microscopy. In particular, films that are mixtures of oleic acid/sodium oleate and fructose are studied as proxies of atmospheric aerosols. The authors explore the different types of nanostructures formed within such internal mixtures as a function of fructose mass fraction and as a function of relative humidity. They also derive hygroscopicity values for the different nanostructures and investigate the reactivity of oleic acid with ozone within these. They find both hygroscopicity and reactivity to be impacted by the nanostructure of the mixtures.

Overall, I feel that the data presented here provides new insights into the physical chemical properties of atmospheric aerosols, a topic that is of interest for the readership of ACP. However, I would like the authors to address the points below, before publication in ACP."

**We are grateful for the positive assessment of our work in terms of suitability for publication in ACP and have addressed the reviewer's comments line-by-line below.**

**"General comments:**

G1: Introduction, L91-112: Please elaborate on your introduction and discussion of the possible three-dimensional nanostructures. It would be good to introduce the nanostructures relevant to this study already here along with e.g. a schematic (as you have them already as very small insets in your Fig. 1). This would greatly help the reader to visualize and distinguish the various nanostructures discussed throughout the text. I also encourage the authors to add clear labels and terms to each of the nanostructures and use the same terminology throughout the text, to make it easier for a reader to follow."

We have added an additional figure (new Figure 1) in the Introduction seciton, showing the different nanostructures formed, and the following new text was also inserted (page 3, line 97): "These are three-dimensional nanostructures which can vary from spherical and cylindrical micelles to bicontinuous networks and bilayers. The spherical and cylindrical micelles can exist with "normal" (oil in water) or "inverse" (water in oil) curvature; the latter are the class formed by the systems in this paper (Pfrang et al., 2017). In our studies, the spherical inverse micelles can exist as (disordered) "inverse micellar" phases, or as ordered "close-packed inverse micellar" phases, which may have cubic (Fd3m) or hexagonal (P63/mmc) symmetry. The cylinders typically pack as hexagonal arrays ("inverse hexagonal phase") and the bilayers as "lamellar" stacks. These structures, shown in Figure 1, can be followed by Small-Angle X-ray Scattering (SAXS), which probes the nanometre scale. The close-packed inverse micellar, inverse hexagonal, and lamellar phases all show long-range periodicity, giving rise to Bragg peaks in SAXS patterns whose positions show symmetries and repeat spacings. The (disordered) inverse micellar phase gives a broad hump in SAXS, whose position shifts with micelle size."

"G2: Ozonolysis of coated films, L148: The ozone concentrations used herein are not atmospherically relevant, as acknowledged by the authors. It would be good to reiterate this when discussing Fig. 4 and the timescales therein. Given typical tropospheric ozone concentrations are much lower, what would be the typical atmospheric timescales at which the changes in SAXS pattern take place? In this regard, it would be helpful to consider discussing timescales of ozone exposure, to take the ozone concentration into account."

We have added comments on the ozone concentration used and its relation to atmospheric levels in several places in the manuscript in response to reviewer 1, e.g. (new text is underlined):

"Note that such a high ozone concentration (atmospheric ozone levels rarely exceed 0.1 ppm) was used as it is known that self-assembled semi-solid phases slow the rate of reaction significantly (Pfrang et al., 2017; Milsom et al., 2021a). Therefore, comparatively high ozone concentrations were chosen to be able to observe an oxidative decay during the limited timescale of synchrotron experiments while they are substantially higher than those generally encountered in the atmosphere."

And in the discussion section "It should be noted that we have carried out the ozonolysis experiments presented here only at low humidity and at high ozone levels; the possible implications of this deviation from atmospheric conditions would merit further investigation (noting experimental challenges associated with interfering reactions of highly reactive OH radicals potentially produced in ozonolysis studies at high humidities). Our earlier modelling work (Milsom et al., 2022b) estimated significantly extended half-lives of nanostructured (lamellar-phase) oleic acid for a range of atmospherically relevant film thicknesses and ozone levels (e.g. a half-life increase of ca. 10 days for a 0.75 µm film in ca. 25 ppb ozone; see Fig. 7 in Milsom et al., 2022b)."

**The last sentence of the new text above specifically addresses this reviewer's query on the impact on atmospheric conditions/timescales.**

"G3: The authors studied three different mixtures of oleic acid: sodium oleate:fructose, different in their mixing ratios, or simpler fructose fractions. I am missing some information whether these ratios were chosen to cover typical mixing ratio ranges found in the atmosphere, or for any specific other reasons? Furthermore, the authors note on L198 that "a set of fructose content-dependent nanostructures are possible". Neglecting the RH-dependency of (irreversible; L212) phase changes, can the authors comment on the range of fructose fractions for which they expect each of the three nanostructure types depicted in Fig. 1 (Fd3m phases, hexagonal phases, lamellar phases) to be present? I.e. is the Fd3m structure always expected in such mixtures if the fructose mass fraction is larger than 50% wt?"

The fatty acid/sugar ratios were chosen according to ratios found by Wang et al. (2006) in field studies of real atmospheric aerosols in the Chinese city of Chongqing in winter, where the two main classes of organic components were fatty acids and sugars (3244 and 2799 ng m-3, respectively). This is in line with the approach first reported in Pfrang et al. (2017), hence this is not repeated in the main text here. The composition choices are also informed by practical aspects such as solubility limitations.

As to whether the Fd3m structure is always expected if the fructose mass fraction is larger than 50 wt %: this is unknown - it could be that this or the other symmetry inverse close-packed micellar phase (p63/mmc) is meta-stable. Moreover, at still higher fructose ratios, it may be that the higher curvature induced by the kosmotropic effect produces micelles too small to accommodate all of the fructose, and so some inverted micellar phase (ordered or disordered) would coexist with excess fructose.

"G4: Deriving hygroscopocity values for the different nanostructures is an interesting approach. The  $\kappa\kappa$ -values derived are extremely low, indicating a low water uptake capacity of theses nanostructures. For inorganic materials  $\kappa\kappa$ -values are often between 0.5 to 1.4, i.e. significantly higher than those observed here. The authors should comment on whether subtle differences in hygroscopicity as observed here (Fig. 3) would still be resolvable in aerosols that are internal mixtures of organic and inorganic material, as is often the case for atmospheric particles? Related, for the potential of a particle to act as a CCN, the inorganic fraction is probably considerably more relevant than the nanostructure of the organic fraction. I encourage the authors to include a discussion on this in their Section "Hygroscopicity of observed nanostructures" and when discussing the impact of nanostructure on hygroscopicity (L466-467) and cloud formation (e.g. L534-535) and climate."

Thanks for the comments; while  $\kappa$ -values are low compared with inorganic materials, the values are substantially higher (up to ca. 50 times) than those reported previously for oleic acid (Rickards et al. (2013)); we did add a comment on the typical  $\kappa$ -values of inorganic materials referring to Petters and Kreidenweis (2007); we also added some of the comments made by the reviewer; the additional text reads: "While the  $\kappa$  values reported here are substantially (up to nearly 50 times) above those previously measured for oleic acid (Rickards et al. (2013), it should be noted that  $\kappa$  values for highly-CCN-active salts such as sodium chloride are still higher (between 0.5 and 1.4; Petters and Kreidenweis, 2007), so that the inorganic fraction may be considerably more relevant than the nanostructure of the organic fraction for the potential of a particle to act as a CCN when considering internal mixtures of organic and inorganic materials in atmospheric particles."

**"Specific comments:**

L75: Please add: DOI: 10.1039/c9cp03731d"

**We have added this additional reference proposed by the reviewer.**

"L116: Add punctuation: "in Milsom et al. (2021a)."

**Thanks: we have corrected this typo.**

"L117-119: Can you comment if and to what degree esterification of the oleic acid and the methanol can take place during sample preparation, and how this could impact your results?"

We have seen no evidence of this in any of our published work or experiments: it would manifest as a change in phase behaviour from samples using oleic acid methanol solutions at the start and end of the synchrotron trip. It also seems highly unlikely as fatty acid esterification typically requires high temperatures and/or the addition of an acid catalyst.

"L120: "oleic acid: sodium oleate:fructose)""

**Thanks: we have corrected this typo.**

"Fig. 1a-c: The meaning of the background colormap is unclear; not specified. Also, consider using non-filled, open markers for "inverse micellar" structures in panel d-f, to allow for easier visual distinction."

**Thanks for this suggestion: we have added a colour map scale in the figure and also changed the markers used as proposed.**

"L188: add "(water-absorbing substance)" or "(moisture attracting agent)" or similar to clarify the meaning of humectant."

**We have added "(moisture attracting agent)" as proposed.**

"L189: The relation between "more hydrated" and "lower water-surfactant interactant interfacial curvature" warrants some more detailed explanation considering the discussion following in this paragraph."

We have re-written the relevant paragraph (see also response to reviewer 1 point 6.) and the revised section now reads: "By this logic, larger amounts of fructose should afford more hydrated phases at a given humidity. This can indeed be seen from a comparison of the inverse micellar spacings at high relative humidity (Figures 2 & 3). However, this does not explain the formation of a close-packed inverse micellar phase at 50 wt% fructose vs. inverse hexagonal at 33 wt% fructose, and lamellar at 20 wt% fructose. We suggest that an additional effect is observed during our experiments: the water-surfactant interfacial curvature increases with increasing fructose concentration (Figure 1). This is clear evidence for fructose acting as a kosmotrope – a water-structure-inducing molecule (Kulkarni et al., 2011; Libster et al., 2008; Koynova et al., 1997)."

"L212-214: Is the statement of the irreversibility of the phase change generally true, or could this be dependent on the drying rate, when going from high back to low RH?"

We did not investigate the effect of the drying rate – we only know that the transformation as we observed it did not show reversal. If the initial phase was metastable, we would not expect the transformation to be reversible back to the metastable initial phase, although it may be possible to re-attain it with the right drying rate. If instead the final phase was metastable, it may revert back to the initial phase if left for long enough.

"L220-222: I might be missing something here, but I do not see the coexistence of pink and blue hexagons between 40-60 min in Fig. 1e that you seem to refer to here in the text."

It is not coexistence of differently coloured hexagons, but of pink hexagons (hex 2) and black circles (inverse micellar) – given this apparent lack of clarity, we have amended the visual representation in line with the proposal of reviewer 1 (black circles were changed to empty circles in the revised figure) and we have also added the following text to the figure caption: "The additional phases coexisting with the (disordered) inverse micellar phase are the cubic close-packed inverse micellar (Fd3m) phase (a,d,g); two different inverse hexagonal phases (b,e,h); and the lamellar phase (c,f,i)."

"L227: Your Fig. 1d shows red squares (i.e. Fd3m structures) for ~175-200 min, i.e. at RH < 90%, please clarify."

It is correct that the red squares persist after reducing humidity from 90% to 70% after ca. 175 min and they only disappear after ca. 200 min – this may indicate that this Fd3m phase is metastable at 70%; the phase disappears at the next RH reduction step to 60% RH.

"L236: This inverse micellar phase does only seem to appear after around 50 min in Fig. 1e, despite the RH being constant for the previous ~20 min. Why is that?"

From Figure 2 (b) scattering at low q (at the bottom of the plot) can be seen after 25 minutes; however, it does not have a well-defined maximum, so does not allow us to determine a meaningful d-spacing for the phase, which may be a polydisperse mixture of inverse micelles of different sizes.

"L297: Add: "The hygroscopicity of the disordered...""

**Thanks: we have corrected this typo.**

"Fig. 4: Figure 4 and the discussion of it (L344 onwards) warrant some improvements. I interpret the solid black arrows in Fig. 4 as indicators of general trends, which peaks disappear and appear, but a clear explanation is missing. Consider adding some specific labels to the individual peaks. Also, the same-colored line corresponds to different times during ozonolysis across the different panels, which is a bit misleading. Having a consistent color code (map) throughout the panels could help with that."

Thanks for the comment; we have explained the meaning of the arrows now: these indicate general trends in peak evolution with time as correctly interpreted by the reviewer; the additional text in the figure captions reads *"The black arrows indicate the progression of different peaks from ordered phases with time as a visual guide."* The colour scheme was chosen on purpose to indicate the four different stages with initial (blue), final (red) and two intermediate states (orange and green) for each plot with consistent colours; we believe this is clearer than matching colours to specific times especially as the time steps are different for the plots that need to be compared (e.g. 10 min, 16 min and 18 min steps only appear once, so that we would need to add extra colours which would cause confusion in our view); all essential information is now clearly displayed and explained in the figure together with the revised caption.

"L382: I am unclear why you specify "(dry) lamellar" here, since your ozonolysis was only per- formed under dry conditions (L344). Related, would you expect the stated trend to be different at different RH conditions?"

Thanks for the query; as we mentioned in the text, the dry lamellar phase is likely more crystalline, with close-packed chains. It is possible that at different RH conditions, a hydrated sample could form the fluid lamellar phase, allowing greater mobility and accessibility of the hydrocarbon chains.

"Fig. 6: a-d: An explanation of the color map(s) is missing and should be added. Also, adding horizontal lines to highlight the peaks at 1650 cm-1 and 1442 cm-1 in panels a and b could help to guide the eye."

Thanks for your suggestions; we have added colour legends in the revised figure as proposed; we did not include horizontal lines to the colour maps as it looked too messy and did not help visual clarity.

"L475-480: This is a very interesting finding in my eyes. Is there a way to highlight this aspect already in the abstract of the manuscript? Right now, it is only mentioned here and on L426."

Thanks for your appreciation of the importance of this finding; this is now highlighted in the additional statement in the conclusions "We have also shown that ozonolysis can induce the formation of a new intermediate molecular arrangement, demonstrating the possibility that self-assembly could be induced by the chemical reaction of these atmospheric components with ozone. This, in combination with humidity-induced phase changes, suggests a dynamic aerosol phase state which is dependent on the molecular arrangement of the surfactant molecules."

"L532: "... with significant impacts on air quality and climate": I suggest toning this down a little bit here, as your work shows the fundamental effects of nanostructure on water uptake and reactivity. These parameters in turn affect the particles impacts on air quality and climate, but an assessment of these effects is not done here."

We agree with the reviewer's comment and added the following caveats to the end of the conclusions section and removed the previous statement in the conclusions on "significant impacts on (i) urban air quality [...] and (ii) climate"; the last paragraph now reads: "Our work demonstrated the fundamental effects of nanostructure on water uptake and reactivity. While these parameters in turn affect the particles' impacts on air quality and climate, a direct assessment of these effects is not within the scope of the work presented here."

---

## Author Response (AR2)

We are grateful for the editor's comments and have amended title and abstract accordingly; the new title now reads "**Experimental observation of** the impact of nanostructure on hygroscopicity and reactivity of fatty acid atmospheric aerosol proxies" and the new abstract now reads "Atmospheric aerosol hygroscopicity and reactivity play key roles in determining an aerosol's fate and are strongly affected by its composition and physical properties. Fatty acids are surfactants commonly found in organic aerosol emissions. They form a wide range of different nanostructures dependent on water content and mixture composition. In this study we follow nano-structural changes in mixtures frequently found in urban organic aerosol emissions, i.e. oleic acid, sodium oleate and fructose, during humidity change and exposure to the atmospheric oxidant ozone. Addition of fructose altered the nanostructure by inducing molecular arrangements with increased surfactant-water interface curvature. Small-Angle X-ray Scattering (SAXS) was employed for the first time to derive the hygroscopicity of each nanostructure**, thus addressing a current gap in knowledge** by measuring time- and humidity-resolved changes in nano-structural parameters. We found that hygroscopicity is directly linked to the specific nanostructure and is dependent on the nanostructure geometry. Reaction with ozone revealed a clear nanostructure-reactivity trend, with notable differences between the individual nanostructures investigated. Simultaneous Raman microscopy complementing the SAXS studies revealed the persistence of oleic acid even after extensive oxidation. Our findings demonstrate that self-assembly of fatty acid nanostructures can significantly impact two key atmospheric aerosol processes: water uptake and chemical reactivity, thus directly affecting the atmospheric lifetime of these materials. This could have significant impacts on both urban air quality (e.g. protecting harmful urban emissions from atmospheric degradation and therefore enabling their long-range transport), and climate (e.g. affecting cloud formation), with implications for human health and wellbeing."

We also checked Fig. 4 with a Color Blindness Simulator and could not detect any issues (it was created with the default python (matplotlib) colour scheme which is designed to accommodate colourblind people).